# Rational design of universal immunotherapy for TfR1-tropic arenaviruses

Hadas Cohen-Dvashi[1], Ron Amon[2], Krystle N. Agans [3,4], Robert W. Cross[3,4], Aliza Borenstein-Katz[1], Mathieu Mateo [5], Sylvain Baize[5], Vered Padler-Karavani[2], Thomas W. Geisbert[3,4] & Ron Diskin [1*]

Certain arenaviruses that circulate in rodent populations can cause life-threatening hemorrhagic fevers when they infect humans. Due to their efficient transmission, arenaviruses pose a severe risk for outbreaks and might be exploited as biological weapons. Effective countermeasures against these viruses are highly desired. Ideally, a single remedy would be effective against many or even all the pathogenic viruses in this family. However, despite the fact that all pathogenic arenaviruses from South America utilize transferrin receptor 1 (TfR1) as a cellular receptor, their viral glycoproteins are highly diversified, impeding efforts to isolate cross-neutralizing antibodies. Here we address this problem using a rational design approach to target TfR1-tropic arenaviruses with high potency and breadth. The pan-reactive molecule is highly effective against all arenaviruses that were tested, offering a universal therapeutic approach. Our design scheme avoids the shortcomings of previous immunoadhesins and can be used to combat other zoonotic pathogens.

---

[1] Department of Structural Biology, Weizmann Institute of Science, Rehovot 7610001, Israel. [2] Department of Cell Research and Immunology, Tel Aviv University, Tel Aviv 69978, Israel. [3] Galveston National Laboratory, University of Texas Medical Branch, Galveston, TX 77555, USA. [4] Department of Microbiology and Immunology, University of Texas Medical Branch, Galveston, TX 77555, USA. [5] Unité de Biologie des Infections Virales Emergentes, Institut Pasteur, Centre International de Recherche en Infectiologie (INSERM, CNRS, ENS Lyon, Université Lyon I), Lyon, France. *email: ron.diskin@weizmann.ac.il

Viral hemorrhagic fevers are a major global health problem. The Ebola virus disease crisis of 2013 to 2016 emphasized the importance of developing and stockpiling effective countermeasures before the onset of deadly outbreaks. Immunotherapeutic agents hold great promise as countermeasures against deadly viruses[1–3]. The *Arenaviridae* is a virus family that encompasses several hemorrhagic fever viruses. Several Arenaviruses that propagate in rodent reservoirs (aka mammarenaviruses) may cause acute and sometimes lethal illness upon infecting humans[4,5]. "New World" (NW) mammarenaviruses, prevalent in the South and North Americas, are classified into four different clades[6,7]. Pathogenic NW mammarenaviruses include the clade-B Machupo (MACV), Junín (JUNV), Guanarito (GTOV), and Sabiá (SBAV) viruses, which are endemic to Bolivia, Argentina, Venezuela, and Brazil, respectively[6,8–10]. In addition, genetically close isolates of the North American clade-A/B Whitewater Arroyo virus (WWAV) may also be pathogenic to humans[11,12]. All these viruses utilize TfR1 as their cell entry receptor[13], and the ability to utilize human-TfR1 (hTfR1) distinguishes them from non-pathogenic viral species[11,14–16].

The surfaces of arenaviruses are coated with trimeric class-I glycoproteins containing a GP1 subunit that adopts a unique fold[17] and mediates receptor recognition[18]. Neutralizing monoclonal antibodies (mAbs) against JUNV that target the receptor-binding site on GP1, as well as sera from JUNV-convalescent patients, generally do not cross-neutralize other NW arenaviruses[19], due to structural variations in the receptor-binding sites[19–21]. Although cross-neutralization against MACV was observed with a vaccine-elicited anti-JUNV antibody[22], neutralization of additional NW mammarenaviruses by this antibody was not reported. Since neutralizing mAbs against JUNV can rescue animals from a lethal challenge[23], it would be beneficial to extend this approach and to generate analogous reagents that could potently target each of the pathogenic members of this family. Better yet would be a single reagent that neutralizes all pathogenic NW arenaviruses regardless of their structural variation.

Immunoadhesins are engineered molecules consisting of protein decoys that mimic viral cellular receptors fused to Fc portion of antibodies. Following a successful demonstration of using receptors as decoys[24], this strategy was explored for potential use in combating HIV-1[25]. In principle, immunoadhesins should have remarkable breadth toward a complete class of viruses that share the same receptor tropism. Despite great promise, however, attempts to use human-derived receptors as immunoadhesins have so far failed[26], and no anti-viral immunoadhesin has yet been approved for clinical use. A basic conceptual flaw that may account for this failure relates to the fact that, despite having excellent breadth, these reagents generally suffer from low potency. The limited potency is due to the mechanism of action of the immunoadhesins: they compete in a stoichiometric fashion with the native receptors, which are generally highly abundant in the human host. Under such conditions, a very high dose of immunoadhesin, which may not be clinically achievable, must be used to obtain good therapeutic activity. It would therefore be advantageous to construct immunotherapeutic agents that not only have the breadth of immunoadhesins but also a clinically relevant potency.

Here we are constructing a highly potent and broad-spectrum immunotherapeutic agent to widely target TfR1-tropic mammarenaviruses. We are utilizing host-derived TfR1 ortholog as part of our immunoadhesin to achieve high potency. Our immunoadhesin is effectively neutralizing a wide range of murine leukemia virus (MLV)-pseudotyped viruses as well as live infectious mammarenaviruses. It is further mediating Fc-effector functions and hence provides an attractive approach for fighting infections by TfR1-tropic mammarenaviruses. The approach that we are using here could potentially be utilized to target other zoonotic viruses.

## Results

**Design of a soluble TfR1 mimetic**. As a potential broadly reactive immunotherapy against NW pathogenic mammarenaviruses, we designed a TfR1 mimetic that blocks the GP1 receptor-binding sites. TfR1 is a large homodimeric type-II transmembrane glycoprotein (Fig. 1a) with a butterfly-like shape[27,28]. Three subdomains constitute each subunit of the extracellular region of TfR1 (Fig. 1b): a helical domain that mediates dimerization, a protease-like domain, and an apical domain that is inserted between two β-strands of the protease-like domain (Fig. 1b, c). The binding site for the TfR1-tropic mammarenaviruses is in the apical domain[28], which is not involved in the main physiological roles of TfR1 in binding transferrin[29] or hereditary hemochromatosis protein[30], and only mediates the interaction of TfR1 with ferritin[31]. Therefore, a mimetic of the apical domain should have only minimal interference with the normal functions of TfR1.

We hypothesized that zoonotic viruses like the NW mammarenaviruses are best adapted to use the cellular receptors of their natural animal reservoirs rather than the human cellular receptors. Rodent TfR1s can serve as efficient entry receptors to various arenaviruses, including non-pathogenic species[11], but only a subset of NW mammarenaviruses can use hTfR1[11,14–16]. We demonstrated that TfR1 from *Neotoma albigula* (White-throated woodrat) has remarkably higher affinities to a couple of viral GP1s compared with hTfR1[21]. Also, White-throated woodrat TfR1 (wwTfR1) can efficiently serve as an entry receptor for various mammarenaviruses[11]. Thus, we based our apical-domain design on wwTfR1 instead of on hTfR1 to achieve high-affinity targeting of pathogenic arenaviruses. Using receptor orthologs derived from animal reservoirs instead of human-derived receptors marks a conceptual change for designing immunoadhesins.

We introduced several modifications to the wwTfR1-apical domain to allow it to be produced as a stand-alone protein. These modifications included the elimination of a long loop (residues 301–326) from the apical domain (Fig. 1b, c) and the mutation of several hydrophobic residues at the interface between the apical- and protease-like domains to increase hydrophilicity (Fig. 1c, d). Lastly, we incorporated two cysteine residues at the termini of the stand-alone apical domain (sAD) to increase stability (Fig. 1c). This design allowed the expression of sAD to serve as a receptor-binding site competitor.

The designed sAD is a soluble, folded, and stable protein. After affinity purification of sAD using a His$_6$-tag at its C′ terminus, a single peak corresponding to monomeric protein was obtained using size exclusion chromatography (Fig. 2a), indicating that sAD is monodisperse. Furthermore, the circular dichroism (CD) spectrum for sAD was characteristic of a folded protein, with a negative peak at a wavelength of 222 nm, indicating helical content (Fig. 2b). Following the CD signal at 222 nm, we monitored the thermal stability of sAD (Fig. 2c). A complex biphasic melting curve was observed, so we did not attempt to fit a model to these data to derive a precise melting point. Nevertheless, sAD is completely thermostable up to 55 °C, and we can estimate the $T_M$ to be ~65 °C. Thus, sAD is sufficiently soluble and stable to be considered for therapeutic use.

**TfR1-derived sAD binds pathogenic mammarenaviral glycoproteins**. To neutralize TfR1-tropic viruses, sAD must bind their GP1 domains. To evaluate binding, we first constructed a series of

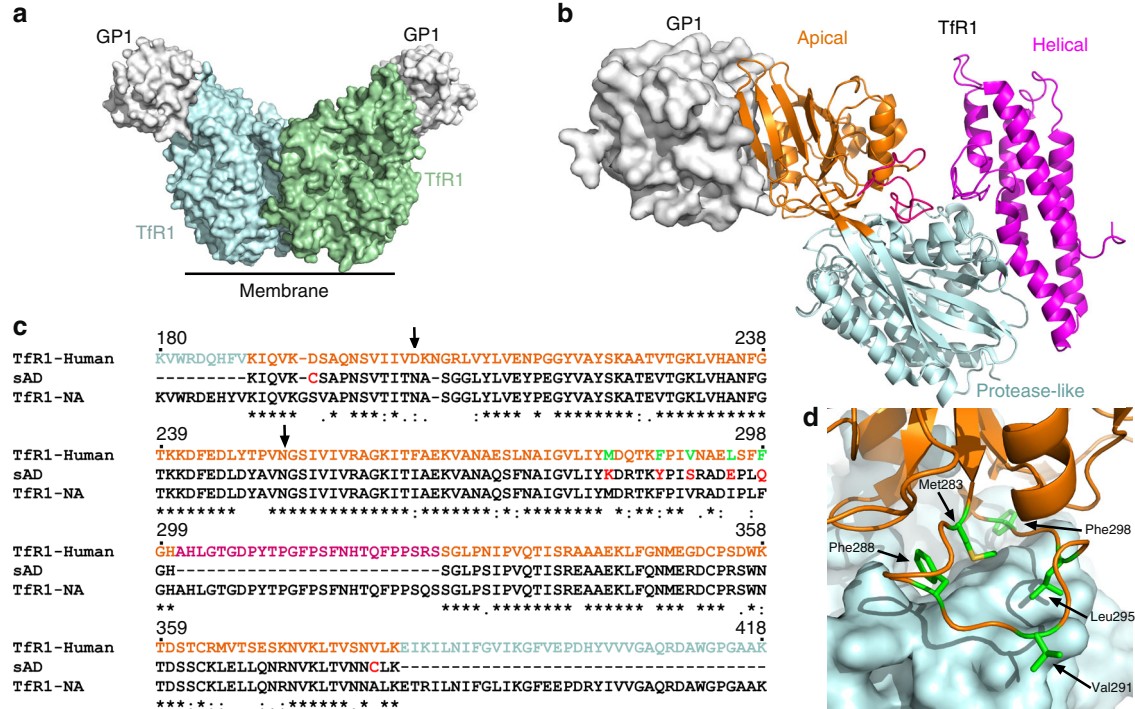

**Fig. 1 Design of a soluble apical domain from TfR1. a** Overview of the TfR1/GP1 complex structure (PDB ID: 3KAS). Two GP1 molecules from MACV (gray) bound to the dimeric human-TfR1 (light-blue and green). **b** The apical domain of TfR1 (orange) is sequence imbedded within in the protease-like domain (light-blue), and together with the helical dimerization domain (magenta) makes one complete copy of the TfR1 molecule. **c** Sequence alignment of human-TfR1, White-throated woodrat (WW) TfR1, and sAD. The numbering scheme follows the human-TfR1 numbering, and the sequence of the human-TfR1 is colored according to the color scheme as in **b**, **d**. The potential N-linked glycosylation sites are indicated with black arrows. **d** A close-up view of the hydrophobic interface of the apical domain (orange) and the protease-like domain (light-blue). The hydrophobic residues that were mutated in sAD are shown in green.

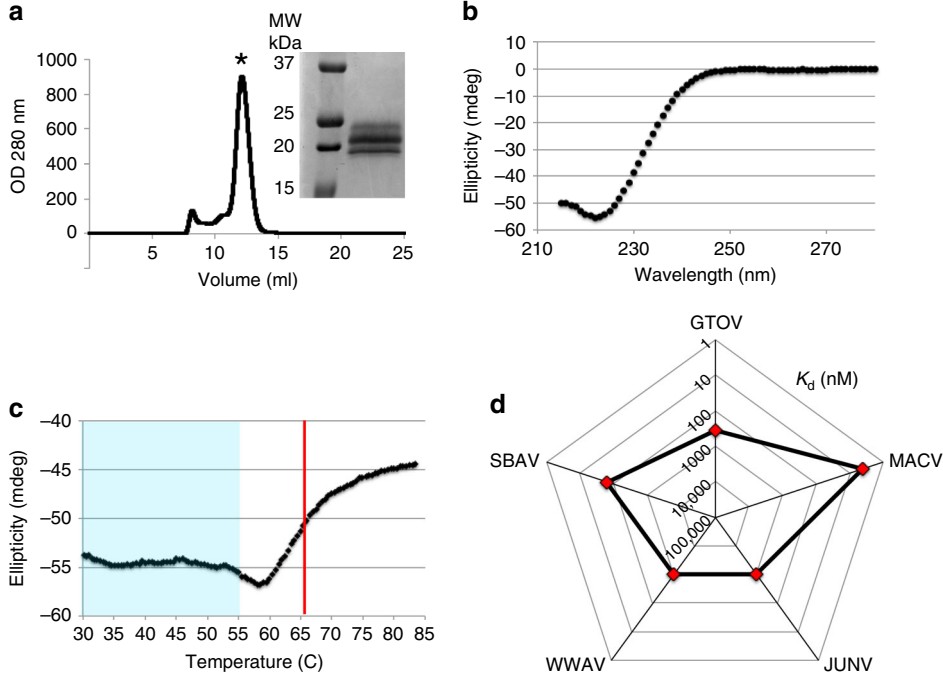

**Fig. 2 The designed apical domain makes a soluble and stable protein that effectively binds a range of GP1 domains. a** Size exclusion chromatography profile of the soluble apical domain after affinity purification demonstrates a predominant monodisperse monomeric peak (mark with an asterisk). Inset shows SDS-PAGE analysis of purified sAD. Three glycoforms are evident. **b** Representative circular dichroism spectrum of the sAD demonstrates a well-folded protein. **c** Thermal denaturation of sAD. The circular dichroism signal was monitored at 222 nm while ramping temperature. The sAD is stable until 55 °C (light-blue shaded region), with an estimated $T_M$ of ~65 °C (red line). **d** A spider graph showing the dissociation constants ($K_d$) between sAD and the indicated GP1 domains from clades B and A/B mammarenaviruses, as measured using SPR.

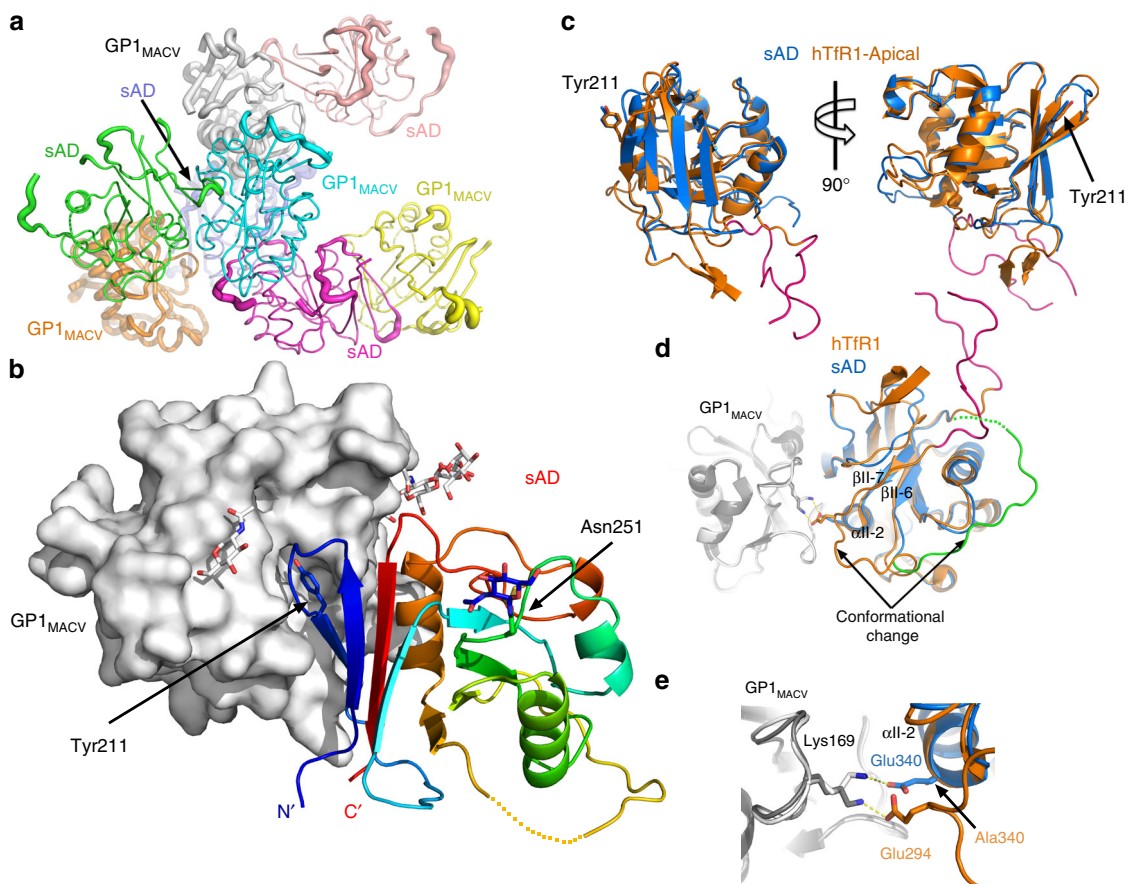

**Fig. 3 Isolated sAD retains a similar structure to the apical domain of TfR1. a** The asymmetric unit contains four copies of the sAD/GP1$_{MACV}$ complex. Each of the eight chains is shown using a unique color. The chains are rendered as tubes with radii proportional to the *B*-factor. The protein pairs in the asymmetric unit differ in quality of the electron density; some have a low *B*-factors (e.g., green/cyan), while others are less defined and hence have higher *B*-factor (e.g., purple/orange). Regardless, the sAD/GP1 interface is identical in all pairs. **b** Crystal structure of sAD in complex with GP1$_{MACV}$. The GP1 domain is shown using surface representation (white), and sAD is presented as a ribbon diagram in rainbow colors from the N′ terminus (blue) to the C′ terminus (red). N-linked glycans and Tyr211 of sAD are shown using sticks. **c** The sAD adopts the same overall structure as the apical domain of hTfR1. A ribbon diagram showing the apical domain of hTfR1 (PDB ID: 3KAS) in orange superimposed on sAD, shown in blue. The right view is rotated 90° with respect to the view on the left. Tyr211, a central residue at the interface with GP1, is shown. Residues 301–326 of hTfR1, which were omitted in sAD, are colored pink. **d** A ribbon diagram showing the structure of the complex between sAD and GP1$_{MACV}$, in blue and white, respectively, superimposed on the hTfR1/GP1$_{MACV}$ complex (PDB ID: 3KAS), colored orange and gray, respectively. The long loop that connects strands βII-6 and βII-7 changes position in sAD compared to hTfR1 and is highlighted in green (sAD). This loop in hTfR1 originally included residues 301–326 (pink), which were eliminated from sAD. **e** Similar superimposition as in **d**, showing the charge–charge interaction between GP1$_{MACV}$ and sAD or hTfR1. The negatively charged Glu294 of hTfR1 forms a salt bridge with Lys169 from GP1. In the case of sAD, Glu340 from αII-2, which replaces an alanine residue of hTfR1, projects in the same direction as Glu294 of hTfR1. This Glu340 of sAD forms a similar salt bridge with Lys169 of GP1$_{MACV}$.

GP1 domains fused at their C′ termini to Fc portions of antibodies. We included GP1 domains from JUNV, MACV, GTOV, and SBAV, which are the major pathogenic mammarenaviruses from clade-B, and further included WWAV as a TfR1-tropic clade-A/B representative. We performed single-cycle kinetics experiments using surface plasmon resonance (SPR) and measured the dissociation constants ($K_d$) of sAD to the various representative GP1 domains, in a configuration that allowed monovalent binding (Supplementary Fig. 1). The sAD reagent effectively binds all GP1 domains, with $K_d$ values ranging from 4 nM for MACV to 1 μM for JUNV and WWAV (Fig. 2d).

To verify the binding mode of sAD to GP1, we crystallized and solved the structure of GP1$_{MACV}$ in complex with sAD to 2.7 Å resolution (Supplementary Table 1 and Supplementary Fig. 2). Crystals belonged to a tetragonal space group ($P4_322$) with four copies of sAD/GP1$_{MACV}$ in the asymmetric unit (Fig. 3a). The designed sAD forms a complex with GP1$_{MACV}$ (Fig. 3b) in a similar fashion to hTfR1[28], while maintaining the overall structure of the hTfR1-apical domain (Fig. 3c). Of two potential N-linked glycosylation sites of sAD (Fig. 1c), we observed electron density and hence modeled a glycan only at the Asn251 position (Fig. 3b). Most of the important interactions that GP1$_{MACV}$ makes with hTfR1 are also formed with sAD, including the key interaction with Tyr211 (Fig. 3b). We did observe some structural differences; however, the long loop that connects parallel strands βII-6 and βII-7 of sAD, which was mutated and shortened (Fig. 1b–d), changed its conformation compared with hTfR1 (Fig. 3d). In the case of hTfR1[28], Glu294 from this loop forms a salt bridge with Lys169 of GP1$_{MACV}$ (Fig. 3e), and in the case of sAD an alternative salt bridge with Lys169 is formed with Glu340 from αII-2 instead (Fig. 3e). Overall, sAD mostly preserves the native structure of the TfR1-apical domain and shows remarkably broad reactivity with GP1s from NW mammarenaviruses.

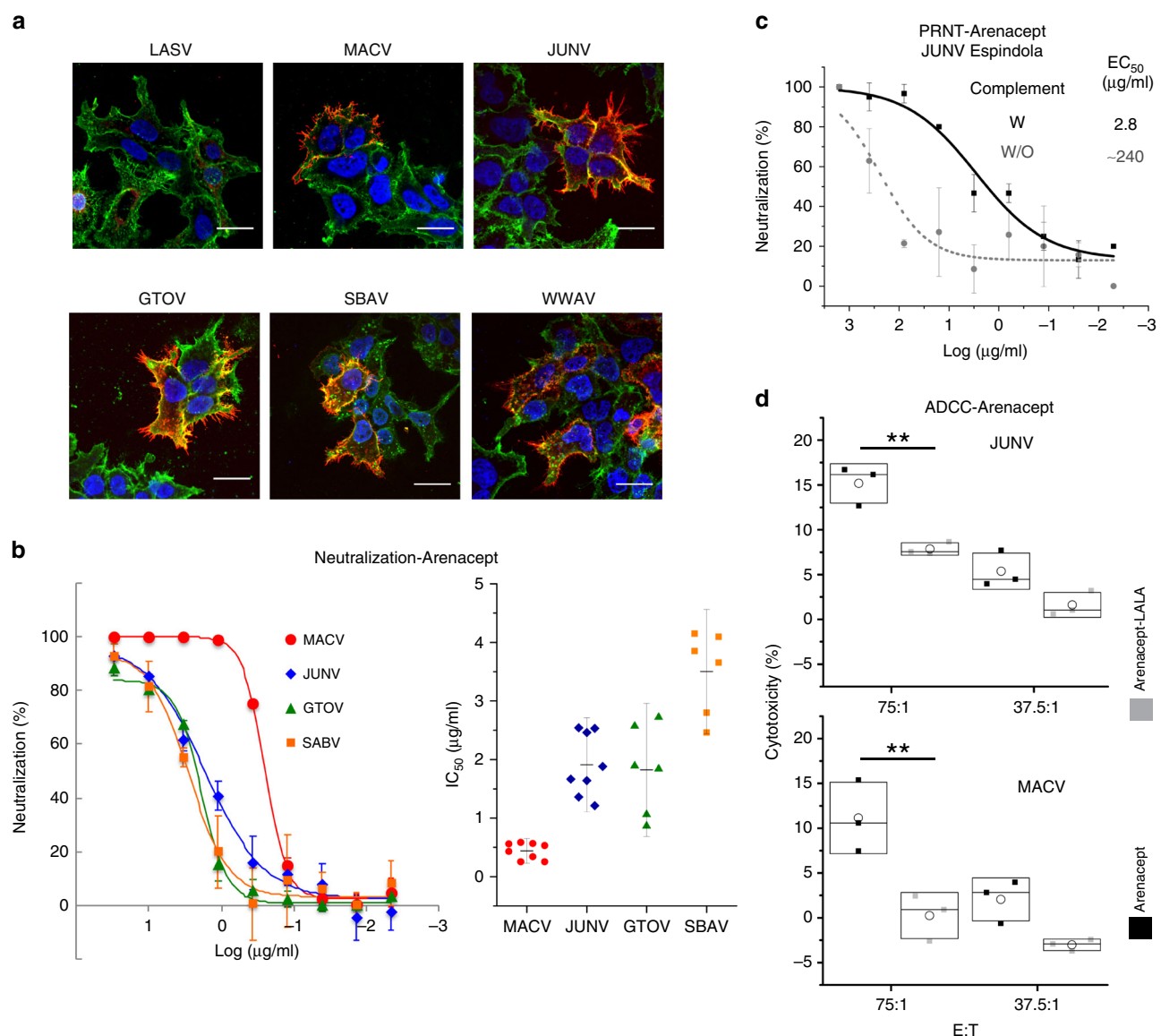

**Fig. 4 Arenacept is biologically active against pathogenic viruses. a** Confocal fluorescence imaging of HEK293, transiently transfected with genes encoding GPCs from the indicated viruses and stained with Arenacept. Nuclei were stained with DAPI (blue), membranes were stained with wheat germ agglutinin (green), and Arenacept was visualized using fluorescent anti-human Ab (red). Scale bars represent 20 µm. **b** Neutralization of pseudotyped viruses. Graphs show representative neutralization of pseudoviruses that bear spike complexes from the indicated viruses (left). Infection was monitored by a luciferase reporter gene in a stable HEK293 cell line that overexpresses hTfR1. Error bars show standard deviations from technical replicates. Measured $IC_{50}$ values are shown on the right. Each dot corresponds to an independent neutralization experiment. Bars indicate standard deviations, and horizontal lines indicate averaged values. The titers of pseudoviruses used for these experiments were not predetermined. **c** Plaque reduction neutralization test using live Espindola strain JUNV. One hundred plaque-forming units of JUNV-Espindola were overlaid on Vero cells in the presence of Arenacept at the indicated concentrations. Virus-only control was used as a reference for calculating neutralization. Error bars show standard deviations from technical replicates. **d** Arenacept promotes ADCC in cells expressing the spike complexes of JUNV and MACV. HEK293A target cells "T" that express the spike complexes of the viruses were incubated with PBMC effector cells "E" at increasing ratios in the presence of 10 µg/ml of Arenacept or Arenacept-LALA. Release of lactate dehydrogenase was used to measure cytotoxicity. ADCC activities against spike-presenting target cells were compared to control cells transfected with an irrelevant plasmid. Graphs are representatives of three independent experiments. Boxes indicate standard deviations that were derived from $n = 3$ technical replicates. Median and average values are indicated with horizontal lines and open circles, respectively. Statistical significance was calculated using two-way ANOVA (**$P < 0.01$). For all relevant panels, source data are provided as a source data file.

**TfR1-based immunoadhesin targets pathogenic NW mammarenaviruses**. Having demonstrated the functionality of sAD in GP1 binding, we converted it to an immunoadhesin and designated it "Arenacept." Specifically, the sAD C′ terminus was fused to an Fc portion of IgG1 in a configuration that links two sAD domains and allows both of them to bind virus simultaneously to achieve avidity. We first tested whether Arenacept recognizes the native spike complexes of TfR1-tropic viruses. Confocal fluorescence imaging revealed that Arenacept binds the native spike complexes of MACV, JUNV, GTOV, SBAV, and WWAV (Fig. 4a). This recognition is specific, as the spike complex of the non-TfR1-tropic Old World Lassa mammarenavirus is not recognized by Arenacept (Fig. 4a). Next, we examined whether Arenacept could neutralize pseudotyped viruses bearing the spike

complexes from the pathogenic viruses. We generated MLV-based gammaretroviral-pseudotyped viruses that deliver a gene encoding luciferase upon entering cells and monitored the reduction in transduction in the presence of Arenacept (Fig. 4b). For this assay, we used a HEK293T cell line that overexpresses TfR1. These cells are significantly more susceptible to the pseudotyped viruses and hence provide much stronger signal for cell entry. Applying Arenacept effectively neutralized MACV, JUNV, GTOV, and SBAV with mean calculated half-maximal inhibitory concentration ($IC_{50}$) values of 0.4–3.4 μg/ml (Fig. 4b). WWAV-pseudotyped viruses do not efficiently infect HEK293 cells, so neutralization could not be evaluated. Introducing into Arenacept a Y211A mutation, which eliminates a critical contact with GP1 (Fig. 3b), abrogated neutralization activity against JUNV (Supplementary Fig. 3), indicating that Arenacept preserves the same binding mode observed for sAD (Fig. 3b). The similar, low $IC_{50}$ values of Arenacept for the various viruses (Fig. 3b) compared with the marked differences in affinities of sAD to the GP1s (Fig. 2d) imply that avidity plays a critical role for neutralization. Indeed, the monovalent sAD has a lower neutralization capacity compared with Arenacept (Supplementary Fig. 4). Thus, Arenacept utilizes avidity and successfully neutralizes all four pseudotyped viruses bearing glycoprotein complexes (GPCs) from the pathogenic clade-B viruses tested.

After obtaining a promising reagent that is capable of neutralizing a diverse set of TfR1-tropic mammarenaviruses, we tested whether the use of TfR1 ortholog derived from hosts that are natural reservoirs is indeed required to achieve potent neutralization. For that, we constructed a new sAD based on hTfR1 and fused it to an Fc portion of IgG1 (hTfR1-Fc) to create a human version of Arenacept. We compared side by side the neutralization potentials of hTfR1-Fc and Arenacept using pseudotyped viruses (Supplementary Fig. 5). Consistent with the principle motivating the Arenacept design, hTfR1-Fc displays orders of magnitudes weaker neutralization compared to Arenacept. This observation indicates that the use of orthologs derived from hosts, which are natural reservoirs, for immunoadhesins is a promising approach for achieving potent reagents and overcomes a major limitation of analogous reagents constructed using human receptor fragments.

Next, we evaluated the risk of potential competition between Arenacept and TfR1 in binding ferritin. Comparing the structure of hTfR1 in complex with ferritin[31] to that of Arenacept indicates that two important polar contacts between hTfR1 and ferritin will not form between ferritin and Arenacept due to a tyrosine residue at position 215 of Arenacept that is a natural wwTfR1 residue, which substitutes an asparagine residue of hTfR1 (Supplementary Fig. 6a). Moreover, the bigger side chain of Tyr215 may sterically interfere with binding to ferritin (Supplementary Fig. 6a). To test that hypothesis, we immobilized Arenacept and hTfR1-Fc on SPR sensor chips and used ferritin as an analyte in single-cycle kinetics experiments (Supplementary Fig. 6b). Since ferritin is a 24-mer, this experimental configuration enables ultra-high avidity and only provides apparent macroscopic binding affinities. Indeed, using this binding geometry ferritin readily and irreversibly binds to hTfR1-Fc (Supplementary Fig. 6b). In contrast, and despite the ultra-high avidity, ferritin binds but dissociates from Arenacept (Supplementary Fig. 6b). Hence, the interaction of ferritin with Arenacept is substantially weaker than with hTfR1.

**Arenacept can neutralize live infectious viruses**. To validate the neutralization capacity of Arenacept against live infectious viruses, we performed a plaque reduction neutralization test (PRNT) in a BSL-4 facility. As a stringent test, we used JUNV, which was not the most sensitive virus to Arenacept as indicated by the

neutralization assays with the pseudotyped viruses (Fig. 4b). Live infectious Espindola strain of JUNV was applied to Vero cells in the presence of Arenacept at various concentrations with or without 5% of guinea pig complement, and the reduction in the number of plaques was monitored 5 days later. In the presence of complement, Arenacept effectively neutralized JUNV-Espindola with a half-maximal effective concentration ($EC_{50}$) ($PRN_{50}$) value of 2.8 μg/ml (Fig. 4c). This value agrees with the $IC_{50}$ value that was obtained with the JUNV-pseudotyped viruses (Fig. 4b). Interestingly, in the absence of complement, the neutralization capacity of Arenacept was significantly reduced (Fig. 4c). Hence, Arenacept is effective against live infectious JUNV in a mechanism that is enhanced by complement-dependent cytotoxicity (CDC).

**Arenacept induces Fc-mediated cellular cytotoxicity**. We next examined whether Arenacept can further eliminate infected cells by inducing antibody-dependent cellular cytotoxicity (ADCC). We transiently expressed the spike complexes of MACV and JUNV in HEK293 cells, applied peripheral blood mononuclear cells from healthy donors to the transfected HEK293 cells, and monitored cell-killing activity in the presence of Arenacept (Fig. 4d). As a control, we further measured cell-killing activity mediated by a modified version of Arenacept carrying two mutations in the Fc region, L234A, and L235A (Arenacept-LALA) that causes reduced affinity to Fcγ receptors and hence reduced capacity to promote ADCC[32]. We observed a clear increase in cytotoxicity as a function of the ratio of effector to target cells when applying Arenacept, but significantly less so when applying Arenacept-LALA (Fig. 4d). Interestingly, Arenacept induced more robust ADCC in the case of JUNV compared to MACV, but in both cases the ADCC activity was significant. Thus, Arenacept has the potential to promote clearance of infected cells in addition to directly neutralizing viruses.

**Enhancing Arenacept through rational design**. Following the promising indications for the ability of Arenacept to neutralize viruses and to recruit Fc-mediated immune functions, we explored the possibility of further enhancing its potency. Our original sAD construct bears two putative N-linked glycosylation sites (Fig. 1c), which are both partially glycosylated as indicated by the three glycoforms that appear in the sodium dodecyl sulfate-polyacrylamide gel electrophoresis (SDS-PAGE) analysis of sAD (Fig. 2a). In the crystal structure of sAD/$GP1_{MACV}$, we observed electron density for N-acetylglucosamine only at the Asn251 glycosylation site of sAD (Fig. 3b) and not near Asn204. This observation implies that the sAD/$GP1_{MACV}$ complex may selectively form with sAD molecules that do not bear a glycan at position Asn204. In the sAD/$GP1_{MACV}$ structure, Asn204 of sAD is in close proximity to $GP1_{MACV}$, suggesting that an N-acetylglucosamine attached to this residue will sterically prevent the formation of a complex (Fig. 5a). As a consequence, Arenacept molecules that have either one or two sAD arms that are glycosylated at position Asn204 will not be able to utilize avidity or will be completely inert, respectively. We thus introduced an S206A mutation to the sAD portion of Arenacept (referred to as Arenacept-M1) to eliminate the N-X-T/S glycosylation motif. Using side-by-side comparisons with pseudotyped viruses, Arenacept-M1 displays enhanced neutralization capacities against most of the viruses that we tested (Fig. 5b).

Since CDC is important for efficient neutralization of live infectious JUNV (Fig. 4c), we further introduced an E430G mutation, which slightly increases the formation of complement-activating Fc hexamers[33], into the Fc portion of Arenacept-M1 (referred to as Arenacept-M3). We repeated the PRNT using live

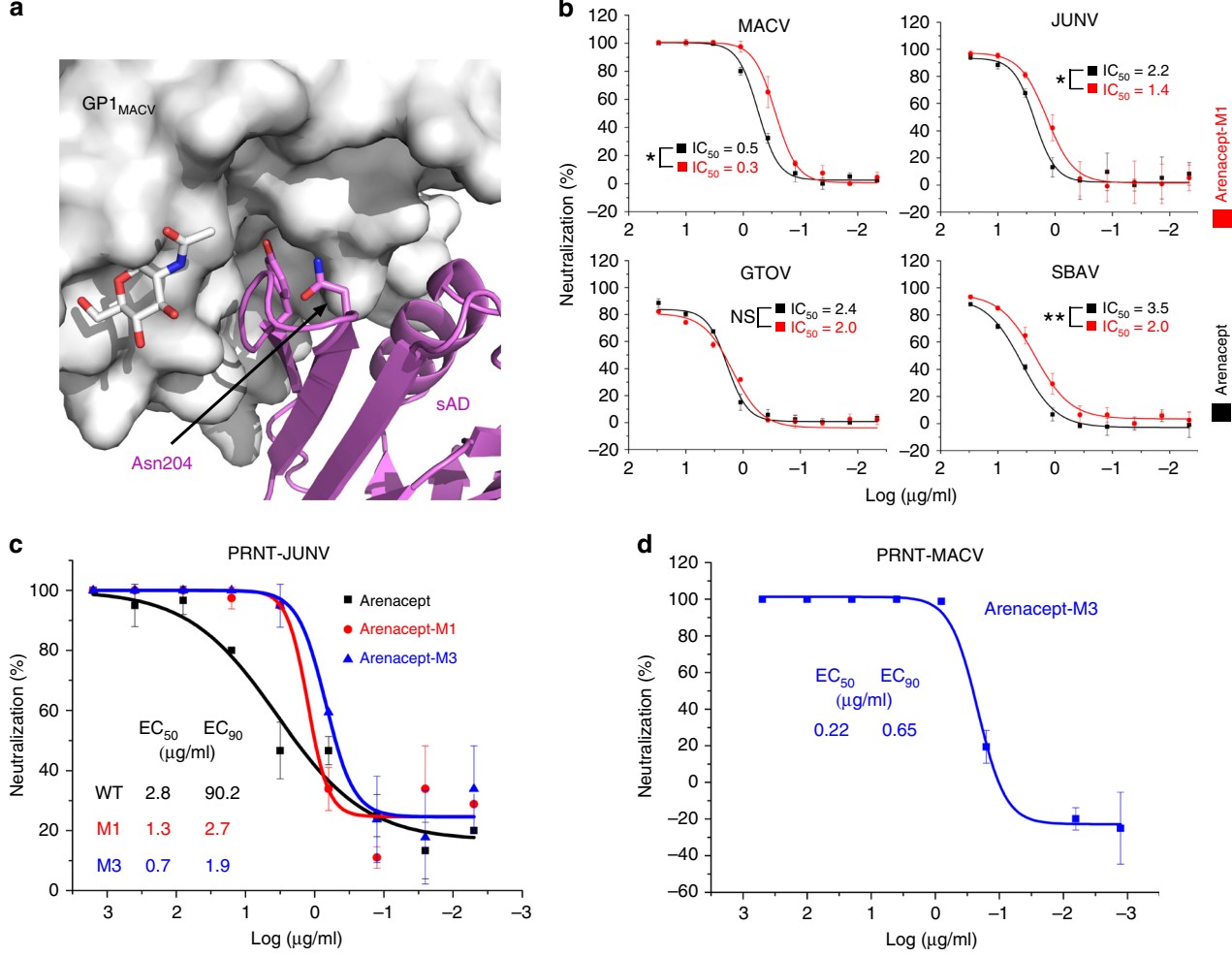

**Fig. 5 Augmenting Arenacept via rational design. a** A glycan attached to Asn204 of sAD would prevent binding to GP1. The sAD/GP1$_{MACV}$ complex is shown in a gray surface and in magenta ribbon for GP1$_{MACV}$ and sAD, respectively. Asn204 of sAD, which is in proximity to the GP1 surface, is indicated. **b** Neutralization of the indicated pseudotyped viruses with Arenacept (black curves) and Arenacept-M1 (N204A, red curves), representative curves are shown. Differences in IC$_{50}$ values (averages of $n = 3$, $n = 5$, $n = 3$, and $n = 3$ independent repeats for MACV, JUNV, SBAV, and GTOV, respectively) that are statistically significant (two-tailed Student's $t$ test) are indicated with *$p < 0.05$ or **$p < 0.005$. Error bars represent standard deviations of technical repeats **c** Plaque reduction neutralization test using live Espindola strain JUNV and Arenacept (WT, black curve), Arenacept-M1 (M1, red curve), and Arenacept-M3 (M3: N204A + E430G, blue curve) in the presence of complement. For each variant, the EC$_{50}$ and EC$_{90}$ values are indicated. Error bars represent standard deviations. For this experiment, we used two technical repeats. **d** Plaque reduction neutralization test using live Carvallo strain MACV and Arenacept-M3 in the presence of complement. The EC$_{50}$ and EC$_{90}$ values are indicated. The neutralization curve was generated by aggregating data from three technical repeats. Error bars represent standard deviations. For all relevant panels, source data are provided as a source data file.

infectious JUNV with both Arenacept-M1 and Arenacept-M3 (Fig. 5c). Remarkably, eliminating the Asn204 glycosylation site substantially altered the profile of the neutralization curves (Fig. 5c), with calculated hill slopes of 2.9 and 2.1 for Arenacept-M1 and Arenacept-M3, respectively vs. 0.7 for Arenacept. Such steeper slopes indicate a marked increase in cooperativity. This change results in modestly improved EC$_{50}$ values but a substantial improvement of EC$_{90}$ values for both Arenacept-M1 and Arenacept-M3 (Fig. 5c). Arenacept-M3, which has a stronger tendency to activate CDC, displays even better EC$_{50}$ and EC$_{90}$ values compared to Arenacept-M1, further indicating the importance of complement activation for neutralizing live infectious JUNV. Having such a potent reagent, we further tested if Arenacept-M3 has the capacity to target other live infectious mammarenavirus as implied from the neutralization experiments with the pseudotyped viruses (Figs. 4b and 5c). We evaluated the neutralization capacity of Arenacept-M3 using PRNT against infectious MACV (Carvallo strain). We found that Arenacept-M3

very potently neutralizes MACV in the presence of complement with an EC$_{90}$ value below 1 μg/ml (Fig. 5d), establishing Arenacept-M3 as a highly potent, broad-spectrum reagent.

**Mechanism of complement-enhanced neutralization.** The marked enhancement of neutralization by the addition of complement in the PRNT against JUNV (Fig. 4c) and to some degree against MACV (Supplementary Fig. 7) may result from several distinct mechanisms: complement may directly enhance neutralization of viruses by increasing the valence of Arenacept and subsequently disrupting the viruses or sterically occluding them from cells. Alternatively, residual complement and Arenacept in media may target infected cells that display the viral spike complex on their surfaces. To gain insights into the mechanism of complement enhancement, we performed neutralization experiments using the MLV-pseudotyped viruses in the presence or absence of complement. In this system, the viruses do not

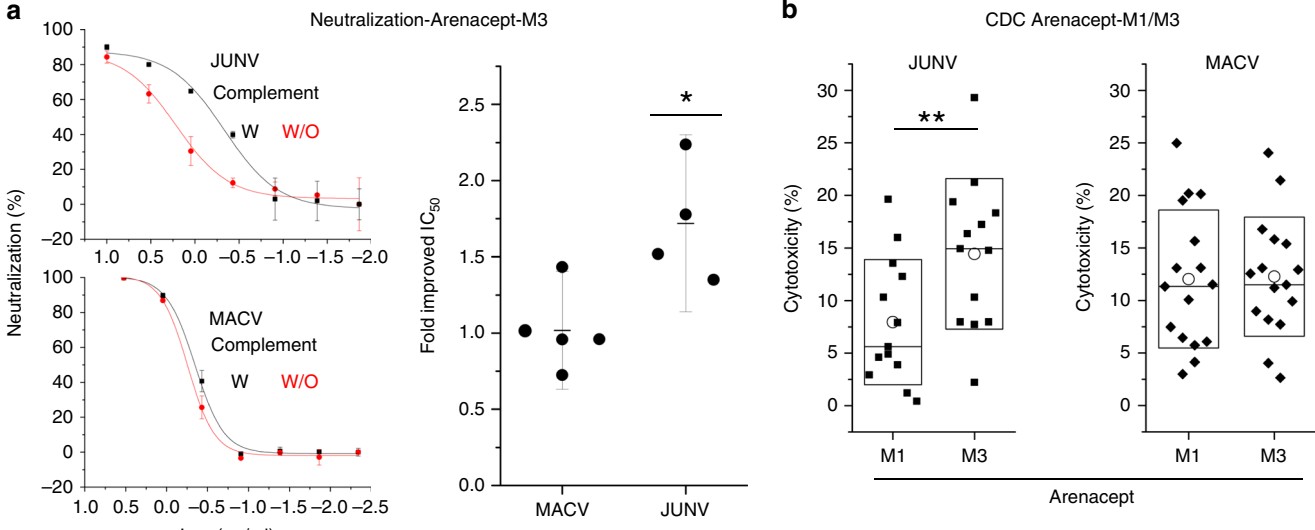

**Fig. 6 Complement mediated enhancement of Arenacept. a** Complement augments direct neutralization of Arenacept-M3. Neutralization of JUNV- and MACV-pseudotyped viruses was tested with or without 2.5% complement. Representative neutralization curves are shown (left). Error bars indicate standard deviations of technical replicates. Fold improvement of $IC_{50}$ values in the presence of 2.5% complement is indicated (right). Each dot represents the fold change between a pair of independent neutralization experiments, with or without complement. Improvements in $IC_{50}$ values are not significant for MACV ($n = 5$ independent experiments), but are significant for JUNV ($n = 4$ independent experiments, * indicates $p < 0.05$, Student's two-tailed $t$ test). **b** Arenacept activates complement. Cytotoxicity mediated by complement activation on HEK293T cells that transiently display the spike complex of JUNV (left) or of MACV (right). Cell killing was determined at Arenacept concentration of 10 μg/ml Arenacept-M1 or Arenacept-M3, as indicated. The difference in cytotoxicity between Arenacept-M1 and Arenacept-M3 in the case of JUNV is significant ($n = 13$ technical repeats, ** indicates $p < 0.005$, Student's two-tailed $t$ test). These are representative experiments out of three independent repeats. Boxes indicate standard deviations of technical repeats. Median and average values are indicated with horizontal lines and open circles, respectively. For all relevant panels, source data are provided as a source data file.

replicate in cells, and enhancement of neutralization could result only from direct action on the pseudoviruses themselves. Indeed, complement significantly augmented Arenacept-M3 neutralization of JUNV-pseudotyped viruses (Fig. 6a). Interestingly, the addition of complement did not augment Arenacept-M3 in the case of MACV-pseudotyped viruses (Fig. 6a). Next, we tested whether Arenacept is able to mediate CDC of cells that transiently display the spike complexes of JUNV or of MACV. In both cases, we observed robust killing mediated by Arenacept (Fig. 6b). In the case of JUNV, we further noticed a significant increase in CDC by using Arenacept-M3 instead of Arenacept-M1 (Fig. 6b). Thus, complement directly potentiates neutralization of viruses by Arenacept, and it is also effectively recruited by Arenacept to kill infected cells.

## Discussion

The development of Arenacept offers a promising immunotherapeutic approach for combating infections by the notorious pathogenic clade-B mammarenaviruses, which pose a health threat for millions of people in endemic regions and so far has had very limited options for treatment. Targeting the apical domain of TfR1 to block infection was previously suggested[28], and subsequently demonstrated using an antibody[34] and an aptamer[35]. However, this approach requires systemic-wide blocking of all TfR1 molecules, which might not be readily achieved, may cause undesired side effects, and cannot induce clearance of infected cells. The apical domain of TfR1, on the other hand, can directly target viruses and even promote clearance of infected cells by binding to the viral spike complex on their surfaces. Our design of sAD resulted in a thermostable protein, an advantageous property that would be instrumental for reducing manufacturing costs and for distributing it in regions with poor clinical and logistical infrastructures.

Using pseudoviruses we found Arenacept to be effective against a set of distinct NW species, and using PRNT in BSL-4 labs was found to be effective against live infectious JUNV and MACV. Interestingly, in the absence of a complement, Arenacept was less potent against live viruses compared to the pseudotyped viruses. Such differences between pseudoviruses and live viruses were previously noted[22,36] and may result from differences in the density and distribution of spike complexes between the pesudoviruses and live viruses, which may ultimately modulate the neutralization potency. The addition of complement greatly enhances the efficacy of Arenacept in a combined mechanism that enhances direct neutralization and likely further targets infected cells by CDC. In the presence of a functional immune system in vivo, ADCC and perhaps additional mechanisms such as opsonization and reduced viral egress by tethering may further promote the function of Arenacept.

Other immunotherapeutic reagents that can target NW mammarenaviruses were previously reported. mAbs against JUNV that were first isolated in mice[37] were shown to protect against a lethal challenge in an animal model[23]. Additional mAbs from vaccinated individuals were shown to neutralize live JUNV, and to be effective against MACV as well[22]. These studies used different strains and experimental conditions for measuring neutralization, so it is difficult to directly compare the potency of such mAbs to Arenacept. Nevertheless, against infectious viruses and in the presence of complement, Arenacept-M3 seems to be slightly less potent compared with some mAbs[22] but more potent compared with others[23]. The most fundamental and important difference between Arenacept and other mAbs, however, is the superior breadth of Arenacept.

Immunoadhesins are promising therapeutic reagents that were successfully introduced into clinical use to treat various, non-viral pathological conditions[38–42]. Due to their inherent breadth and their natural resistance to viral escape mutations, immunoadhesins

still hold great promise as anti-viral therapeutic agents as well. Using receptor orthologs from animal host reservoirs of zoonotic viruses has the potential to result in much more potent immunoadhesins than using human receptor orthologs, as we demonstrated here. One issue of seeming concern is the possibility that immunoadhesins made from animal-derived proteins would be immunogenic. Rodent and human-TfR1 are highly similar in sequence (Fig. 1c), but for the Arenacept model to be generalizable, high sequence similarity should not be assumed. It is important to note that the clinical use of reagents such as Arenacept would not require chronic administration, but rather treatment during life-threatening acute viral diseases. In such a scenario, any putative immunogenicity is not likely to pose a significant concern. This methodology of using host-derived receptors has a potential to be effective against other zoonotic viruses, and hence to become a general strategy for constructing broad-spectrum and potent immunotherapeutic agents.

## Methods

**Construction of expression vectors**. Codon-optimized forms of MACV, JUNV, GTOV, and SBAV GPC genes were chemically synthesized (Genscript), according to their UniProt sequences, as follows: JUNV (O10428), GTOV (Q8AYW1), and SBAV (H6V7J2). Genes encoding WWAV and MACV GPC were a kind gift from Dr. Hyeryun Choe. All GPCs were subcloned into the pcDNA3.1 expression vector, using *Bam*HI–*Not*I restriction sites. The gene encoding sAD was chemically synthesized (Genscript), and was subcloned into pACgp67b vector for production in insect cells using *Bam*HI–*Not*I restriction sites. GP1$_{JUNV}$–Fc, GP1$_{MACV}$–Fc, GP1$_{GTOV}$–Fc, GP1$_{SBAV}$–Fc and GP1$_{WWAV}$–Fc, and sAD-Fc (Arenacept) fusion proteins were generated following a similar protocol as we previously used[43]. Briefly, the core regions of the GP1s (residues 85–244, 80–223, 77–246, 80–223, and 80–231 for MACV, GTOV, JUNV, WWAV, and SBAV, respectively) were cloned upstream of a human Fc region derived from IgG1 and downstream of a signal peptide using *Bam*HI–*Kpn*I restriction sites. Mutated variant Y211A of Arenacept was generated by PCR using Kapa HiFi DNA polymerase (Kapa Biosystems) according to the QuikChange site-directed mutagenesis manual. Human transferrin receptor-encoding vector hTfR1-pENTR221 was obtained from the Weizmann Institute Forscheimer plasmid bank and was subcloned into pQXIP using *Bam*HI–*Not*I restriction sites. A gene encoding wwTfR1 was chemically synthesized (Genscript).

**Protein expression and purification**. To express and purify the complex of sAD with GP1MACV for structural studies, we used the same methodologies as used for producing GP1LASV[43]. Briefly, the two proteins were co-expressed as secreted proteins using the baculovirus system in Tni (Trichoplusia ni) cells (Expression Systems). Media were collected and buffer exchanged to TBS (20 mM Tris-HCl, pH 8.0, 150 mM sodium chloride) using a tangential flow filtration system (Millipore). The complex was captured using a HiTrap IMAC FF Ni$^{+2}$ (GE Healthcare) affinity column followed by size exclusion chromatography purification with a Superdex 75 10/300 column (GE Healthcare). Fc-fused GP1s (GP1$_{JUNV}$–Fc, GP1$_{MACV}$–Fc, GP1$_{GTOV}$–Fc, GP1$_{SBAV}$–Fc, and GP1$_{WWAV}$–Fc), Arenacept, and the various His-TfR1s were expressed in suspension-HEK293F cells (Gibco) grown in FreeStyle media (Gibco). Transfections were done using linear 25 kDa polyethylenimine (PEI) (Polysciences) at 1 mg of plasmid DNA per 1 L of culture at a cell density of 10$^6$/ml. Media were collected after 5 days and supplemented with 0.02% (w/v) sodium azide and phenylmethylsulfonyl fluoride. Fusion proteins were isolated using protein-A or HiTrap IMAC FF Ni$^{+2}$ (GE Healthcare) affinity columns.

**SPR measurements**. Binding of sAD to GP1$_{JUNV}$–Fc, GP1$_{MACV}$–Fc, GP1$_{GTOV}$–Fc, GP1$_{SBAV}$–Fc, and GP1$_{WWAV}$–Fc fusion proteins was measured using a Biacore T200 instrument (GE Healthcare). Fusion proteins were first immobilized at a coupling density of ~500 resonance units (RU) on a series S sensor chip protein A (GE Healthcare) in TBS and 0.02% sodium azide buffer. One of the four flow cells on the sensor chip was coupled with GP1$_{LASV}$–Fc to serve as a blank. sAD was then injected at 5, 50, 250, 500, and 1000 nM concentrations, at a flow rate of 80 μl/min. Single-cycle kinetics was performed for the binding assay. The sensor chip was regenerated using 10 mM glycine-HCl pH 1.5 buffer. The binding of hTfR1 and wwTfR1 to GP1$_{JUNV}$–Fc and GP1$_{MACV}$–Fc was similarly measured, at TfR1 concentrations of 500, 250, 125, 12.5, and 1.25 nM.

**In vitro neutralization assays**. Pseudoviral particles of MACV, JUNV, GTOV, and SBAV were produced as previously described[44], except for the use of pLXIN-Luc as the reporter gene (pLXIN-Luc was a gift from Alice Wong, Addgene plasmid # 60683). Media containing pseudoviruses were concentrated 10× by PEG precipitation. For that, the viral-containing media were supplemented with PEG 6000 (Sigma) in phosphate-buffered saline (PBS) to a final concentration of 8%

(w/v). Following incubation of 18 h at 4 °C, viruses were pelleted by centrifugation at 10,000 × *g* for 20 min. Pellets of viruses were resuspended in cell culture media.

For generating a stable cell line that overexpresses hTfR, HEK293T (ATCC) cells were transfected with the hTfR-pQXIP vector. At 48 h post transfection, media were replaced and supplemented with 2 μg/ml puromycin for selection. Cells were grown in the presence of antibiotics for 1 week. Resistant colonies of stable cells were collected and cultured in the presence of puromycin to form a polyclonal cell line.

For neutralization assays, hTfR-stable HEK293T were seeded on poly-L-lysine pre-coated white, chimney 96-well plates (Greiner Bio-One). Cells were left to adhere for 2 h, followed by the addition of 10× concentrated pseudoviruses, which were pre-incubated with threefold descending concentrations of either Arenacept or sAD, with or without the addition of 2.5% rabbit complement (Cedarlane). Cells were washed from viruses 18 h post infection, and luminescence from the activity of luciferase was measured 48 h post infection using a TECAN infinite M200 pro plate reader after applying Bright-Glo reagent (Promega) on cells.

**Cell staining and fluorescence-microscopy imaging**. HEK293T cells were seeded on poly-L-lysine pre-coated coverslips in 24-well plates and transfected with different GPCs using PEI reagent. At 24 h post transfection cells were incubated for 5 min with 1 μg/ml Arenacept diluted in cell culture media, fixed with pre-warmed 3.7% (v/v) formaldehyde (paraformaldehyde) solution in PBS, and blocked with 3% (w/v) bovine serum albumin in PBS. Cells were stained with Cy3-conjugated anti-Human Fc (Jackson Laboratories, dilution 1:500, catalog # 109-165-008) and fluorescein isothiocyanate-conjugated wheat germ agglutinin (Thermo Fisher 1:200, catalog # W11261). Cells were imaged at ×100 magnification using an Olympus IX83 microscope coupled to a Yokogawa CSU-W1 spinning disc confocal scanner. Images were processed using ImageJ.

**CD measurements**. Stock solution of 10 mg/ml sAD in 20 mM Tris-HCl (pH 8.0), 150 mM sodium chloride was diluted 1:40 in 150 mM sodium chloride solution for recording CD spectra using a Chirascan-plus ACD spectrometer. For determining temperature stability of the protein, CD spectra at a wavelength of 222 nm were measured at temperatures ranging between 30 °C and 85 °C (ramping of 0.5°/5 s).

**Crystallization**. Screening for initial crystallization conditions was done with an 8.8 mg/ml stock of the complex sAD/GP1$_{MACV}$, using a Mosquito crystallization robot (TTP Labs). Initial hits were identified using the JCSG-plus screen (Molecular Dimensions) and were optimized manually. Crystals were obtained using sitting drop vapor diffusion in 0.2 M sodium thiocyanate, pH 6.9, 20% (w/v) PEG 3350, and 5% (v/v) MPD. Crystals were then successively cryo-protected using 20% (v/v) MPD in reservoir solution before flash cooling in liquid nitrogen.

**Data collection, structure solution, and refinement**. X-ray diffraction data were collected at the European Synchrotron Radiation Facility (ESRF) beamlines ID30B using a Pilatus 6M-F detector at 100 K. Data to 2.7 Å in a tetragonal space group were collected. HKL2000[45] was used to index, integrate, and scale the data. Phaser[46] was used to obtain a molecular replacement solution using the structure of GP1$_{MACV}$ in complex with the apical domain of hTfR1 (PDB: 3KAS) as a search model. The crystal belonged to a tetragonal *P*4$_3$22 space group and contained four sAD/GP1 complexes in the ASU. The model was manually fitted into electron density maps using Coot[47] and refined using Phenix Refine[46] in an iterative fashion.

**ADCC assays**. For measuring ADCC HEK293A (ATCC) cells were grown to 80–90% confluence in 100 mm plates. Transfection mixtures of 1 ml containing 40 μg/ml 25 kDa PEI (Polysciences) with 8 μg JUNV, MACV, or control plasmid in Dulbecco's modified Eagle's medium (DMEM) were made and incubated for 15 min at room temperature. Two milliliters of media was removed from each culture, and 1 ml transfection mix was added. After 24 h, cells were detached from the plate using 10 mM EDTA. The ability of Arenacept or Arenacept-LALA to promote ADCC was evaluated by measuring lactate dehydrogenase (LDH) release using a LDH Cytotoxicity Detection kit (Roche Applied Science) according to the manufacturer's instructions. HEK293A target (T) cells were transfected with GPC of JUNV, MACV, or an irrelevant vector as control and subsequently incubated at 1 × 10$^5$ cells/ml with or without 10 μg/ml Arenacept or Arenacept-LALA on ice for 1 h. PBMCs were collected from human blood using cell preparation tubes. After extensive washes with PBS, the cells were suspended in RPMI and plated in a 96-well round-bottom plate at different amounts. Subsequently, for each PBMC-containing well, 1 × 10$^4$ target cells were added. We used 1% (v/v) Triton X-100 as maximum-release controls and cells without PBMCs and no Arenacept as low spontaneous release controls. Plates were then incubated for 3 h at 37 °C, and supernatants were collected for LDH release determination. Percentage cytotoxicity was calculated as (cells with Arenacept − cells without Arenacept)/(maximum release − spontaneous).

**Plaque reduction neutralization tests**. For testing neutralization of JUNV, Arenacept was serially diluted fourfold (for the first dilution) and fivefold

(for subsequent dilutions) in infection medium (MEM supplemented with 10% (v/v) fetal bovine serum (FBS), 1% L-Gln; 1% (w/v) Pen/Strep, and 5% (v/v) guinea pig complement (Rockland) starting at 1740 µg/ml. In a BSL-4 lab, JUNV was diluted in infection medium to 2000 pfu/mL. An equal volume of JUNV was added to each of the Arenacept dilutions to achieve a concentration of 100 pfu of JUNV. A virus-only control was incubated with medium alone. The dilutions were then incubated for 1 h at 37 °C in a humidified $CO_2$ atmosphere. Vero E6 cells (ATCC) seeded in 6-well plates to near confluence were infected with the dilutions for 1 h before 0.8% (w/v) agarose was added as an overlay. After 5 days, plaques were visualized by staining the cell monolayer with PBS supplemented with 5% neutral red and 5% FBS (v/v). $EC_{50}$ and $EC_{90}$ were calculated using the 4PL curve fit (Origin).

For testing neutralization of MACV, Arenacept-M3 was serially diluted fivefold in infection medium (DMEM supplemented with 10% FBS; 1% Pen/Strep) with guinea pig complement (2% final) (Rockland) starting at 500 µg/mL. MACV was diluted in infection medium, and an equal volume of MACV was added to each of the Arenacept dilutions to achieve a concentration of 100 pfu of MACV per well. A virus-only control was incubated with medium alone. The dilutions were then incubated for 1 h at 37 °C in a humidified $CO_2$ atmosphere. Vero E6 cells seeded in 12-well plates to near confluence were infected with the dilutions for 1 h before carboxymethylcellulose diluted in DMEM. After 7 days, the number of foci was calculated by focus forming immunodetection using anti-MACV antibodies. $EC_{50}$ and $EC_{90}$ were calculated using the 4PL curve fit (Origin).

**CDC assays**. For measuring CDC, HEK293F cells were grown in suspension to a density of $10^6$ cells/ml and transfected using 40 kDa $PEI^{MAX}$ (Polysciences) with plasmids encoding the GPCs of JUNV and of MACV. Cells were cultured for 48 h post transfection. Cells were plated into 96-well plates at a cell density of 20,000 cells per well. Reaction mixtures containing rabbit complement (Cedarlane) at a final concentration of 2.5% with or without Arenacept at a final concentration of 10 µg/ml were added to wells and incubated for 2 h at 37 °C. To monitor cytotoxicity we followed the release of LDH using a LDH Cytotoxicity Detection kit (Roche Applied Science) according to the manufacturer's instructions. We used 1% Triton X-100 as maximum-release controls, cells with complement but without Arenacept as a spontaneous control, and cells without Arenacept or complement as background control. From all reads we subtracted the background control and calculated percentage cytotoxicity as (cells with Arenacept − spontaneous) × 100/(maximum release − spontaneous).

**Reporting summary**. Further information on research design is available in the Nature Research Reporting Summary linked to this article.

## Data availability

Coordinates and structure factors for sAD/GP1$_{MACV}$ structure are available at the PDB under accession code 6S9J. Raw data underlying Figs. 4b–d, 5b–d, 6a, b and Supplementary Figs. 3–5, 7 are provided as source data file. All other data are available from the corresponding author upon reasonable requests.

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

## Acknowledgements

The genes encoding the GPC from WWAV and MACV were a kind gift of Dr. Hyeryun Choe. Purified H-ferritin was a kind gift of Dr. Beatrice Vallone. Ron Diskin is the incumbent of the Tauro career development chair in biomedical research. We are grateful to Gotthard Guillaume at the ESRF, Grenoble, France for providing assistance in using beamline ID30B. We thank Dr. Yosef Scolnik for his kind help in recording CD spectra, Dr. Emmanuel Levy for granting us access to his confocal microscope, Mr. Daniel J. Deer for technical support, and Dr. Deborah Fass for providing critical comments and suggestions. We thank Miles Carroll and Roger Hewson (Public Health England, Porton Down, UK) for providing the Machupo virus, Carvallo strain. BSL-4 experiments using MACV were supported by the Pasteur-Weizmann joint research program collaborative research grant S-CR18069-02 to M.M. and R.D. BSL-4 operations of the Galveston National Laboratory and Department of Microbiology and Immunology are supported by NIAID/NIH Grant No. UC7AI094660 to T.W.G. The lab of R.D. is supported by research grants from the Enoch Foundation, from the Abramson Family Center for Young Scientists, from Ms. Rudolfine Steindling, and from the I-CORE Program of the Planning and Budgeting Committee and The Israel Science Foundation (grants no. 1775/12 and 682/16).

## Author contributions

R.D. conceived and designed Arenacept; H.C.-D. produced sAD and Arenacept, performed cell imaging, CD, SPR, crystalized sAD/GP1$_{MACV}$, and performed neutralization assays. R.D. solved the structure of sAD/GP1$_{MACV}$; R.A. and V.P.-K. performed ADCC assays and analyzed data; A.K. produced and purified Arenacept in a large scale; K.N.A., R.W.C. and T.W.G. performed PRNT assay on JUNV and analyzed data; M.M. and S.B. performed PRNT assay on MACV and analyzed data; R.D. wrote the manuscript together with all other coauthors.

## Competing interests

R.D. and H.C.-D. have a pending patent application with the Israel patent office No. 253984, titled "IMMUNOTHERAPY AGAINST TRANSFERRIN RECEPTOR 1 (TfR1)-TROPIC ARENAVIRUSES." All other authors declare no competing interests.
