## [Peer Review File · Nature Communications]

Reviewers' comments:

Reviewer #1 (Remarks to the Author):

Cohen-Dvashi, et al. describe the development and analysis of a Transferrin receptor immunoadhesion for use in the treatment of pathogenic New World arenavirus infections. Using the apical domain of TfR1 of the white-throated woodrat (wwTfR1), the authors are able to overcome the challenge of out-competing the abundant expression of human TfR1 on the cell surface. They first optimize and biophysically characterize the apical domain of wwTfR1, termed sAD. The accompanying crystal structure of sAD in complex with the receptor binding subunit of Machupo virus (MACV) confirms the structural fidelity of sAD and demonstrates sAD-MACV GP1 interaction is similar to the human TfR1-MACV interaction. The authors show the sAD immunoadhesion (termed Arencept) is able to neutralize pseudoparticles from numerous pathogenic NW arenaviruses and authentic Junin virus. Using structure-guided engineering, the authors also produce Arencept derivatives with increased potency. This manuscript provides an important first step in the development of a single reagent for the treatment of hTfR1-tropic New World arenaviruses that circumvents the challenges in identifying a broadly cross-reactive and neutralizing antibody.

Specific comments and issues that should be addressed prior to publication:

- 1) There are several areas that could be grammatically improved for clarity and readability. For example, "Demonstrating the potential use of receptors as decoys prompted the design of immunoadhesins to combat HIV-1." is awkward.
- 2) The authors state that "various viral GP1s" have higher affinity for the TfR1 of the white-throated woodrat thus providing the basis for choosing this species' TfR1 for the immunoadhesion. However, the reference cited describes only the interaction between wwTfR1 and WWAV (for which the white-throated woodrat is the host) and MACV. Given the extremely low affinity of wwSAD for both JUNV and WWAV, it would be useful to have some context as to whether further exploration of other species' TfR1 was done as well and whether wwTfR1 is the optimal choice.
- 3) Similarly, the authors state that pathogenic NW arenaviruses are defined by their ability to use hTfR1 and that WWAV might be pathogenic. Yet, further in the results section, they explain that WWAV cannot enter 293T cells over expressing hTfR1. Could the authors please clarify? It would also be interesting to know if the modest changes to hTfR1 reported in the referenced paper would permit WWAV infection and whether arencept could block this infection.
- 4) It would also be useful for the authors to provide a comment on the potency of Arencept (or derivatives) vs the neutralizing antibodies against JUNV and MACV, particularly since JUNV is the most difficult to neutralize by Arencept.

- 5) For the MLV-based pseudovirion experiments, the methods indicate that the viruses are concentrated by 10x, however there is no mention of ensuring the amount of each virus delivered in these experiments is equal. One could certainly expect there to be variation in total titer and that this variation would prohibit comparison of the potency of Arenacept/sAD between each virus. Indeed, the affinity of sAD for SABV is 10-fold greater than for JUNV, yet the neutralization potency is 2x lower for SABV. The authors should at the very least normalize for the amount of virus delivered to the cells or comment on the disconnect between affinity and potency.
- 6) The authors modify Arenacept to specifically increase formation of complement-activating Fc-hexamers but do not show that this mutation actually results in an increase in CDC. The authors should demonstrate that the mutation shows the intended outcome, particularly since JUNV neutralization is enhanced 100-fold in the presence of complement.
- 7) The authors suggest that the deletion of Asn204 in Arenacept enhances cooperativity. If the authors are basing this on the hill slope, they should provide those values or other evidence that the binding is truly cooperative. Otherwise, please remove this statement.
- 8) Panel A in Figure 3 is too small to be meaningful. Can this panel be moved to a supplementary figure and made larger? Otherwise, it adds very little to the overall figure. Further, in the figure legend for this panel the authors say “there are differences in the protein pairs” but do not say what these differences are or if they are important.
- 9) For Figure 5, it would be helpful to have a reminder of what mutations M1 and M3 encode.
- 10) The authors describe “short forms” of wwTfR1 and gpTfR1 (guinea pig?) in the methods but do not reference the use of these constructs in the manuscript itself. Similarly, methods for SPR experiments reference gpTfR1 binding experiments that are not displayed in any figure nor discussed in the manuscript.
- 11) The authors should state the residues encoded for each GP1-Fc construct in the methods.

Reviewer #2 (Remarks to the Author):

Cohen-Dvashi et al. describe the rational design of a biological molecule with broad activity against several pathogenic arenaviruses, most of which have limited treatment options or vaccines. The approach they used to accomplish is novel – it involved generating Fc fusion proteins that present part of the ectodomain of the cellular receptor, TfR1, derived from a rodent host species for an arenavirus, which seems to have a higher affinity to the arenaviral GPs than does human TfR1. While the work is conceptual novel and of interest to the field in general, it falls short in several respects. These include data with infectious Junin virus (JUNV) that show only an effect in the presence of complement. They do not show, however, that complement is required for an effect on the entry of

arenavirus GP pseudotyped viruses. The lack of an effect on infectious virus in the absence of complement puts into question the primary mechanism of action the authors originally envisioned as part of their rational design (e.g., that their constructs inhibit viral entry by serving as a decoy receptor). The data with live infectious virus they present is also only limited to JUNV, the only New World arenavirus for which there is an effective vaccine and an established treatment (passive administration of human plasma). The work would be significantly strengthened, if the authors were to provide data with infectious viruses that represent a broader set of those that currently lack treatment and vaccine options (e.g., Machupo, Guanarito, Sabia, Chapare, and White-Water Arroyo virus), for which such a biomolecule would indeed solve an unmet need.

Major comments:

1. Data with infectious virus are only provided for JUNV. A more comprehensive set of live infectious viruses should be tested to claim that arenacept has antiviral activity against all New world arenaviruses.

2. The authors should provide a detailed explanation in the discussion for why they believe arenacept potentially inhibits viral entry in a pseudotype system but requires complement to be active against infectious JUNV. It appears that the protocol they used in their PRNT assays involved adding the virus/arenacept/complement mixture to cells for one hour prior to adding agarose, which would mean the antibody would remain present even after viral entry into cells. Could arenacept, rather than blocking entry, lead to complement-dependent toxicity of infected cells, or block viral egress, thus preventing the formation of visible plaques? Could the authors perform additional experiments to address this? For example, the authors could repeat the experiment but wash the cells prior to adding the agarose to remove the virus/arenacept/complement mixture in an attempt to tease this out (e.g., if washing the cells abrogates the antiviral activity, their molecule would probably be acting at a step that is downstream of viral entry).

3. Two sentences in the text require modification or clarification because, in light of recent data, they are misleading. "The binding site for the TfR1-tropic mammarenaviruses is in the apical domain, which is not involved in the known physiological roles of TfR1 binding to transferrin or hereditary chromatin protein. Therefore, a mimetic of the apical domain should not interfere with normal function of TfR1" and "The apical domain of TfR1 has no known biological functions and hence makes a potentially safe reagent to be injected into patients as a decoy." The apical domain of TfR1 was recently (and convincingly) shown to bind ferritin in a manner that would compete with binding to arenavirus GP1s (see Montemiglio et al. Nature Communications 2019, for a 3.9A cryo-EM reconstruction of the ferritin-TfR1 complex). The authors should test whether their arenacept construct binds to human ferritin. If it does, this might have implications on therapeutic dosing (whatever amount of ferritin that is present in serum in vivo could in principle chelate their biomolecule and render it inactive).

4. Based on what is written in the methods section, the in vitro experiments with pseudoviruses are seemingly done using HEK293T cell lines that overexpress human TfR1. Numerous other studies using viral pseudotypes with New World arenaviruses use WT HEK293T cells (see, for example, Radoshitzky et al. Nature 2007). The authors should clarify in the main text the rationale for using

HEK293Ts that overexpress TfR1, as opposed to the WT cells. Is entry inhibition on WT HEK293Ts not the same?

5. Fig. 5b. To make a more convincing claim that the activity of arenacept is enhanced by the changes introduced to make arenacept-M1, the authors should provide IC50 values for the curves. Are the differences significant? (For GTOV, in particular, there seems to be no change).

Minor comments:

1. On page 3, second paragraph, the authors write “Some cross-neutralization against MACV was observed with vaccine-elicited anti-JUNV antibodies, but the cross-neutralization potency was modest.” The paper referenced describes one cross-neutralizing monoclonal antibody that has modest cross-neutralizing activity against MACV and another that neutralizes JUNV and MACV with similar IC50 values. It is unclear what the authors, therefore, mean by “the cross-neutralization potency was modest.”
2. On page 7, “The sAD reagent effectively binds all GP1 domains, with KD values ranging from 4 nM for MACV to 1 μ M for JUNV and WWAV.” sAD was derived from the white throated woodrat TfR1 ortholog, which is a suspected host rodent for WWAV – one would, therefore, expect some degree of co-adaptation between this TfR1 ortholog and WWAV GP1. The affinity of sAD for WWAV GP1, however, is weak. The authors should comment on this peculiarity somewhere in the text or in the discussion.
3. On page 16, in the methods section, a construct named “gpTfR1” is mentioned, but it is unclear what this construct is in reference to and whether or not it is included in the work.
4. Fig.1d. It would be useful to label the side chains (with residue positions) for residues that are shown in green.
5. Fig. 3e. Two MACV GP1s seem to be shown. These should be labeled or explained in the figure legend.
6. Fig. 5c. This experiment with infectious virus, based on the text, seems to have been done with complement. If so, this should be explicitly stated in the figure legend.

7. Supplementary Fig.1. In the first panel (GTOV), part of the plot seems to be missing/cut off in the residual plot.

Reviewers' comments:

Reviewer #1 (Remarks to the Author):

Cohen-Dvashi, et al. describe the development and analysis of a Transferrin receptor immunoadhesion for use in the treatment of pathogenic New World arenavirus infections. Using the apical domain of TfR1 of the white-throated woodrat (wwTfR1), the authors are able to overcome the challenge of out-competing the abundant expression of human TfR1 on the cell surface. They first optimize and biophysically characterize the apical domain of wwTfR1, termed sAD. The accompanying crystal structure of sAD in complex with the receptor binding subunit of Machupo virus (MACV) confirms the structural fidelity of sAD and demonstrates sAD-MACV GP1 interaction is similar to the human TfR1-MACV interaction. The authors show the sAD immunoadhesion (termed Arenaccept) is able to neutralize pseudoparticles from numerous pathogenic NW arenaviruses and authentic Junin virus. Using structure-guided engineering, the authors also produce Arenaccept derivatives with increased potency. This manuscript provides an important first step in the development of a single reagent for the treatment of hTfR1-tropic New World arenaviruses that circumvents the challenges in identifying a broadly cross-reactive and neutralizing antibody.

Specific comments and issues that should be addressed prior to publication:

1) There are several areas that could be grammatically improved for clarity and readability. For example, "Demonstrating the potential use of receptors as decoys prompted the design of immunoadhesins to combat HIV-1." is awkward.

We made an effort to improve the clarity and readability of our paper.

2) The authors state that "various viral GP1s" have higher affinity for the TfR1 of the white-throated woodrat thus providing the basis for choosing this species' TfR1 for the immunoadhesion. However, the reference cited describes only the interaction between wwTfR1 and WWAV (for which the white-throated woodrat is the host) and MACV. Given the extremely low affinity of wwSAD for both JUNV and WWAV, it would be useful to have some context as to whether further exploration of other species' TfR1 was done as well and whether wwTfR1 is the optimal choice.

First, we changed the text to clarify that we previously tested only two GP1 domains and not "various GP1s".

Second, our two main reasons for selecting wwTfR1 (as we specify in the manuscript) were: 1) our own observation for the higher affinities of WWAV and MACV to this TfR1, and 2) the report by Hyeryun Choe (Zong M. JVI 2014) that wwTfR1 is a robust receptor for a wide range of NW Arenaviruses (all the ones that were tested in this particular study). Indeed, we did not make a comprehensive survey for finding a putative higher-affinity binder. Nevertheless, from the work of Hyeryun Choe it is clear that while there is a major difference between the human-TfR1 and rodent-derived orthologs comparing their ability to serve as entry receptors, there are only subtle differences comparing different rodent-derived TfR1s across many different NW viruses (see Zong M. JVI 2014 – figure 3 for example). Moreover, although there are indeed large differences in affinities of the wwsAD to the various GP1s, the IC50 values for neutralization of Arenaccept are fairly similar for the various viruses (due to the effect of avidity), hinting that modest improvements in binding affinities will not necessarily result with significant changes in neutralization capacity.

More generally, in this work we managed to design a reagent that we think represents a conceptually novel approach for fighting NW Arenaviruses in particular, and potentially other zoonotic viruses in general. One can certainly envision that development of such a reagent in pre-clinical studies will involve some optimization steps that may also look at alternative TfR1 sequences. Such putative optimization however, will not make according to our opinion, a conceptual change to the results that we have obtained.

3) Similarly, the authors state that pathogenic NW arenaviruses are defined by their ability to use hTfR1 and that WWAV might be pathogenic. Yet, further in the results section, they explain that WWAV cannot enter 293T cells over expressing hTfR1. Could the authors please clarify? It would also be interesting to know if the modest changes to hTfR1 reported in the referenced paper would permit WWAV infection and whether arenaccept could block this infection.

There is some ambiguity in the literature related to the pathogenicity of WWAV. It was first proposed as a pathogenic NW Arenavirus that is related to the death of several individuals (Ensernik M. Science 2000), but then it was shown to not be able to utilize human-TfR1 (Reignier T. JVI 2007). In contrast, an isolate of WWAV termed AV96010151 (Cajimat M.N.B Virus Res 2008) was able (to some extent) to utilize the human-TfR1 (Zong M. JVI 2014). Thus, the pathogenicity of WWAV is still in question and it may be restricted to particular isolates. In our hands, we could not detect entry of WWAV to the human-TfR1 bearing cells. As reviewer # 1 suggests, modest changes to human-TfR1 can indeed convert it into a viable receptor to WWAV. This was already demonstrated and reported by Hyeryun Choe (Zong M. JVI 2014). As for the question whether Arenacept would be able to block the entry of WWAV to cells bearing such rodent-like modified TfR1, one must remember that the prime mechanism that allows Arenacept to neutralize, is its affinity difference toward the GP1s compared to human-TfR1. As we clearly demonstrated with hTfR1-Fc (supplementary figure 4), the efficacy is almost completely disappears when only a simple stoichiometric competition is possible. In this suggested experiment, the affinity of Arenacept to WWAV will be comparable to the affinity of WWAV to the modified rodent-like TfR1, and hence we speculate that the neutralization will be greatly diminished. Regardless, our paper focuses on the development of Arenacept as a putative reagent for combating human infections by NW Arenaviruses. We feel that addressing questions related to entry of viruses into cells that bear rodent-like TfR1 is beyond the scope of this study.

4) It would also be useful for the authors to provide a comment on the potency of Arenacept (or derivatives) vs the neutralizing antibodies against JUNV and MACV, particularly since JUNV is the most difficult to neutralize by Arenacept.

We thank reviewer #1 for this comment. There is very little information about neutralizing mAbs and unfortunately, each of the very few available studies has used different strains and / or different experimental conditions for testing neutralization. Thus, a direct comparison is not very accurate. Nevertheless, we can carefully say that some mAbs (Zeitlin L. et al., PNAS 2016) might be a bit more potent compared to Arenacept, while other mAbs (Clark L.E. et al., Nature Communications, 2018) are clearly much less potent than Arenacept (comparing EC90 values for example). We now added a paragraph to the discussion to address this point.

5) For the MLV-based pseudovirion experiments, the methods indicate that the viruses are concentrated by 10x, however there is no mention of ensuring the amount of each virus delivered in these experiments is equal. One could certainly expect there to be variation in total titer and that this variation would prohibit comparison of the potency of Arenacept/sAD between each virus. Indeed, the affinity of sAD for SABV is 10-fold greater than for JUNV, yet the neutralization potency is 2x lower for SABV. The authors should at the very least normalize for the amount of virus delivered to the cells or comment on the disconnect between affinity and potency.

*Reviewer #1 is absolutely correct that there are variations in the titers of the pseudoviruses that we use. We are fully aware of these variations. However, the IC50 values are intrinsic property of the reagent and the particular virus that is used. That is to say, these IC50 values are titer-independent and the exact same value (within experimental error) should be obtained regardless of the titers used. The titers of the viruses do affect the magnitude of the reporting signal that we record. All of our neutralization experiments were repeated multiple times, using different batches of viral stocks, presumably with different titers. However, as one can clearly see, the individually measured IC50 values (**that we now include in a revised figure 4b**) are nicely clustering, indicating high reproducibility. The fact that we always define maximal infectivity as 100%, allows us to effectively **normalize** all the different experiments as reviewer #1 suggested us to do.*

As for the observed affinities vs. neutralization potencies; mainly due to the effect of avidity, the large gaps in binding affinities (measured using monomeric sAD) are not preserved when comparing neutralization potencies (measured using a bivalent immunoadhesin). Also, although clearly related, neutralization and binding are two very different activities that are governed by very different kinetic constants due to the native trimeric spikes on one hand vs. isolated GP1 domains on the other hand. Hence, we cannot assume that relative differences in binding affinities will be maintained when neutralization potencies are compared.

6) The authors modify Arenacept to specifically increase formation of complement-activating Fc-hexamers but do not show that this mutation actually results in an increase in CDC. The authors should demonstrate that the mutation shows the intended outcome, particularly since JUNV neutralization is enhanced 100-fold in the presence of complement.

Importantly, the E430G mutation is based on published work that already demonstrated its capacity to increase the formation of complement activating hexamers (Diebold C.A, et al. Science 2014). In our data, one can actually see that including this mutation indeed enhances the potency of Arenacept (see M3 vs. M1 in Fig. 5c for example). In the revised manuscript, we now also tested the ability of Arenacept to promote CDC on cells. To address this comment, we included in these experiments both M1 and M3 (see new Figure 6b). Indeed, we see a statistically significant difference in the capacity of Arenacept-M1 to promote CDC compared to Arenacept-M3 against cells displaying the spike complex of JUNV.

7) The authors suggest that the deletion of Asn204 in Arenacept enhances cooperativity. If the authors are basing this on the hill slope, they should provide those values or other evidence that the binding is truly cooperative. Otherwise, please remove this statement.

We now provide the values for the hill coefficients that clearly indicate a gain in cooperativity after eliminating the Asn204 glycosylation site (i.e. a change from 0.7 to 2.1 and 2.9 from WT to M1 and M3, respectively).

8) Panel A in Figure 3 is too small to be meaningful. Can this panel be moved to a supplementary figure and made larger? Otherwise, it adds very little to the overall figure. Further, in the figure legend for this panel the authors say “there are differences in the protein pairs” but do not say what these differences are or if they are important.

As suggested by reviewer #1, we modified this panel to make it clearer. Also, please note that in the figure legend we do specify the differences, which relate to the B-factors of the different chains: “...There are differences between the protein pairs in the asymmetric unit; some are well defined in the electron density, having low B-factors (e.g., green / cyan), others are less defined and hence have higher B-factor (e.g., purple / orange)...”

9) For Figure 5, it would be helpful to have a reminder of what mutations M1 and M3 encode.

We now include a reminder in the figure legend for these mutations

10) The authors describe “short forms” of wwTfR1 and gpTfR1 (guinea pig?) in the methods but do not reference the use of these constructs in the manuscript itself. Similarly, methods for SPR experiments reference gpTfR1 binding experiments that are not displayed in any figure nor discussed in the manuscript.

We now removed this erroneous reference of gpTfR1.

11) The authors should state the residues encoded for each GP1-Fc construct in the methods.

We now state the exact residues for each of the GP1-Fc constructs.

Reviewer #2 (Remarks to the Author):

Cohen-Dvashi et al. describe the rational design of a biological molecule with broad activity against several pathogenic arenaviruses, most of which have limited treatment options or vaccines. The approach they used to accomplish is novel – it involved generating Fc fusion proteins that present part of the ectodomain of the cellular receptor, TfR1, derived from a rodent host species for an arenavirus, which seems to have a higher affinity to the arenaviral GPs than does human TfR1. While the work is conceptual novel and of interest to the field in general, it falls short in several respects. These include data with infectious Junin virus (JUNV) that show only an effect in the presence of complement. They do not show, however, that complement is required for an effect on the entry of arenavirus GP pseudotyped viruses. The lack of an effect on infectious virus in the absence of complement puts into question the primary mechanism of action the authors originally envisioned as part of their rational design (e.g., that their constructs inhibit viral entry by serving as a decoy receptor). The data with live infectious virus they present is also only limited to JUNV, the only New World arenavirus for which there is an effective vaccine and an established treatment (passive administration of human plasma). The work would be significantly strengthened, if the authors were to provide data with infectious viruses that represent a broader set of those that currently lack treatment and vaccine options (e.g., Machupo, Guanarito, Sabia, Chapare, and White-Water Arroyo virus), for which such a biomolecule would indeed solve an unmet need.

Major comments:

1. Data with infectious virus are only provided for JUNV. A more comprehensive set of live infectious viruses should be tested to claim that arenacept has antiviral activity against all New world arenaviruses.

In the revised manuscript we now present data that shows the ability of Arenacept to very effectively neutralize live infectious MACV (new figure 5d). Please note that besides JUNV, MACV is the only other NW mammarenavirus that is currently available to us at the BSL-4 facilities of both UTMB and Institut Pasteur. This new data clearly indicate that Arenacept has the capacity to target more than a single live NW mammarenavirus, and together with our comprehensive evaluations using the pseudoviral system we believe that we have established Arenacept as a broad-spectrum reagent.

2. The authors should provide a detailed explanation in the discussion for why they believe arenacept potently inhibits viral entry in a pseudotype system but requires complement to be active against infectious JUNV.

Others have also noticed that in some cases live viruses tend to be more difficult to neutralize compared to pseudotyped viruses (see Flyak A.I. et al. Cell 2015 for a Filovirus example, and Clark L.E et al. Nature Communications 2018 for an Arenavirus example). Such reports agree with our observation that Arenacept is less potent against live JUNV in the absence of complement. We do not know for sure the reason for this difference but we can speculate that it relates to the distribution and density of spike complexes on the surface of the live viruses compared with the pseudotyped viruses. We now address this point in the discussion.

It appears that the protocol they used in their PRNT assays involved adding the virus/arenacept/complement mixture to cells for one hour prior to adding agarose, which would mean the antibody would remain present even after viral entry into cells. Could arenacept, rather than blocking entry, lead to complement-dependent toxicity of infected cells, or block viral egress, thus preventing the formation of visible plaques?

We thank reviewer #2 for this comment that is now experimentally addressed in our manuscript and adds to our understanding of how Arenacept works.

*We do not think that Arenacept leads to cell toxicity and / or inhibiting viral egress **rather** than blocking entry. Such mechanisms may indeed play a role **in addition** to a direct neutralization the viruses. It is clear from our comprehensive biochemical and structural data that Arenacept is blocking the binding of the TfR1 receptor, and blocking entry is the sole mechanism that is active in the pseudoviral system. Even with live viruses in the absence of complement (JUNV, and in the revised manuscript MACV as well), Arenacept is inhibiting the viruses, clearly indicating that blocking entry via competing with the binding of the receptor is a prime mechanism for neutralization by Arenacept.*

Indeed, the addition of complement greatly enhances the potency of Arenacept. This could be achieved as reviewer #2 suggested by recruiting complement to infected cells. However, this may also be achieved by enhancing the action of Arenacept on viruses prior to cell entry, or by promoting both activities. To test that, we evaluated the ability of complement to modulate neutralization of pseudoviruses on one hand, and to promote CDC on the other (new figure 6).

We now provide new data showing that complement indeed enhances neutralization by Arenacept in the pseudotyped virus system (new Figure 6a). Interestingly, we see a significant effect with JUNV but not with MACV, presumably because there is not much room for improving neutralization of MACV that is highly efficient to begin with. Enhanced neutralization may happen by a direct complement-mediated disruption the viral membrane, by effectively increasing the valency of Arenacept due to the complement-mediated oligomerization or by sterically preventing the viruses from approaching the cell's membrane due to the substantial size of the complement complex.

We next tested the ability of Arenacept to promote CDC on cells displaying the MACV or the JUNV spike complexes (new Figure 6b). In both cases, Arenacept very efficiently activated CDC. We hence think that the observed increase in efficacy in the PRNT assays results from a combined effect of a direct increase in neutralization potency of the viruses that is supplemented by CDC activation on infected cells. Please note that in-vivo, ADCC will also play a role (Figure 4d), and perhaps even other mechanisms like opsonization and blocking viral egress as reviewer #2 suggested. We now discuss the mechanism by which complement enhances the efficacy of Arenacept as well as other potential immune mechanisms in the last paragraph of the results section and in the discussion.

Could the authors perform additional experiments to address this? For example, the authors could repeat the experiment but wash the cells prior to adding the agarose to remove the virus/arenacept/complement mixture in an attempt to tease this out (e.g., if washing the cells abrogates the antiviral activity, their molecule would probably be acting at a step that is downstream of viral entry).

As mentioned above, we addressed this question by testing the ability of complement to enhance neutralization of pseudoviruses and by evaluating the ability of Arenacept to promote CDC (new Figure 6).

3. Two sentences in the text require modification or clarification because, in light of recent data, they are misleading. “The binding site for the TfR1-tropic mammarenaviruses is in the apical domain, which is not involved in the known physiological roles of TfR1 binding to transferrin or hereditary chromatosis protein. Therefore, a mimetic of the apical domain should not interfere with normal function of TfR1” and “The apical domain of TfR1 has no known biological functions and hence makes a potentially safe reagent to be injected into patients as a decoy.” The apical domain of TfR1 was recently (and convincingly) shown to bind ferritin in a manner that would compete with binding to arenavirus GP1s (see Montemiglio et al. Nature Communications 2019, for a 3.9Å cryo-EM reconstruction of the ferritin-TfR1 complex). The authors should test whether their arenacept construct binds to human ferritin. If it does, this might have implications on therapeutic dosing (whatever amount of ferritin that is present in serum in vivo could in principle chelate their biomolecule and render it inactive).

We became aware of this very recent and exciting publication from the lab of Beatrice Vallone only after submitting our manuscript. We now modified the text according to the recent findings. Also, we have obtained human-ferritin from Dr. Vallone and tested the binding of Arenacept as well as of the hTfR1-Fc construct that we have made (new Supplementary Fig. 5). Since the ferritin binding site on TfR1 is not fully conserved between humans and rodents, the binding affinity of ferritin to Arenacept is significantly weaker compared to hTfR1-Fc. We hence think that Arenacept will not be able to efficiently compete with TfR1 on binding to ferritin and we now discuss that in the manuscript.

4. Based on what is written in the methods section, the in vitro experiments with pseudoviruses are seemingly done using HEK293T cell lines that overexpress human TfR1. Numerous other studies using viral pseudotypes with New World arenaviruses use WT HEK293T cells (see, for example, Radoshitzky et al. Nature 2007). The authors should clarify in the main text the rationale for using HEK293Ts that overexpress TfR1, as opposed to the WT cells. Is entry inhibition on WT HEK293Ts not the same?

There is a simple technical reason for using this particular cell line that we have generated. Regular HEK293T cells can certainly be used for detecting cell entry. However, when we measure inhibition of entry we would like to enhance the signal in order to be able to accurately measure low levels of cell entry with highly inhibited viruses (accurately differentiate between 80% and 90% neutralization, for example), in order to get good neutralization curves. With regular HEK293T cells, at these levels of inhibition it is sometimes challenging to record good signal above the background. The TfR1-over expressing cells provide much better signal to noise ratio, and in a sense make a slightly more stringent model for testing neutralization. We now include a short explanation for that in the main-text as reviewer #2 requests.

5. Fig. 5b. To make a more convincing claim that the activity of arenacept is enhanced by the changes introduced to make arenacept-M1, the authors should provide IC50 values for the curves. Are the differences significant? (For GTOV, in particular, there seems to be no change).

We now include the IC50 values as well as a statistical analysis of the differences. Except for GTOV, as reviewer #2 noticed, all other differences are statistically significant.

Minor comments:

1. On page 3, second paragraph, the authors write “Some cross-neutralization against MACV was observed with vaccine-elicited anti-JUNV antibodies, but the cross-neutralization potency was modest.” The paper referenced describes one cross-neutralizing monoclonal antibody that has modest cross-neutralizing activity against MACV and another that neutralizes JUNV and MACV with similar IC50 values. It is unclear what the authors, therefore, mean by “the cross-neutralization potency was modest.”

We modified the text to say "...Neutralizing monoclonal antibodies (mAbs) against JUNV that target the receptor-binding site on GP1, as well as sera from JUNV-convalescent patients generally do not cross-neutralize other NW arenaviruses, due to structural variations in the receptor binding sites 19-21. Nevertheless, cross-neutralization against MACV was observed with a vaccine-elicited anti-JUNV antibody 22, but neutralization of additional NW mammarenaviruses by this antibody was not reported."

2. On page 7, "The sAD reagent effectively binds all GP1 domains, with KD values ranging from 4 nM for MACV to 1 μ M for JUNV and WWAV." sAD was derived from the white throated woodrat TfR1 ortholog, which is a suspected host rodent for WWAV – one would, therefore, expect some degree of co-adaptation between this TfR1 ortholog and WWAV GP1. The affinity of sAD for WWAV GP1, however, is weak. The authors should comment on this peculiarity somewhere in the text or in the discussion.

Indeed, this is an interesting observation but it is not very surprising given what we already know from previous studies. Abraham J. et al. (PLoS Pathogens 2009) showed nicely that some TfR1 orthologs can serve as robust entry receptors for many pathogenic / non-pathogenic NW arenaviruses even though they belong to rodent species that are not necessarily the natural hosts for these viruses. We have shown before (Shimon A. et al., JMB 2017) that receptor selectivity is mainly a function of the specific interactions that each GP1 receptor-binding module has evolved to utilize. Hence, there are GP1s that have higher affinity to TfR1 (like of MACV) and there are others that have low affinities (like WWAV or JUNV). Since TfR1 is fairly conserved, these differences in affinities are generally preserved regardless of the origin of the particular TfR1 ortholog.

Given the abovementioned previous studies, we kindly think that including this discussion, which is not critical to the main message of our manuscript, will increase the clutter without contributing much. If reviewer #2 is not convinced, we could nevertheless add this discussion.

3. On page 16, in the methods section, a construct named "gpTfR1" is mentioned, but it is unclear what this construct is in reference to and whether or not it is included in the work.

We removed this erroneous reference.

4. Fig. 1d. It would be useful to label the side chains (with residue positions) for residues that are shown in green.

We now label the residues.

5. Fig. 3e. Two MACV GP1s seem to be shown. These should be labeled or explained in the figure legend.

We now better explain this figure in the figure legend.

6. Fig. 5c. This experiment with infectious virus, based on the text, seems to have been done with complement. If so, this should be explicitly stated in the figure legend.

We now state that in the figure legend.

7. Supplementary Fig. 1. In the first panel (GTOV), part of the plot seems to be missing/cut off in the residual plot.

We have corrected this residual plot.

REVIEWERS' COMMENTS:

Reviewer #1 (Remarks to the Author):

comment 8:

8) Panel A in Figure 3 is too small to be meaningful. Can this panel be moved to a supplementary figure and made larger? Otherwise, it adds very little to the overall figure. Further, in the figure legend for this panel the authors say "there are differences in the protein pairs" but do not say what these differences

are or if they are important.

Response: "As suggested by reviewer #1, we modified this panel to make it clearer. Also, please note that in the figure legend we do specify the differences, which relate to the B-factors of the different chains:

"...There are differences between the protein pairs in the asymmetric unit; some are well defined in the electron density, having low B-factors (e.g., green / cyan), others are less defined and hence have higher B-factor (e.g., purple / orange)..."

The confusion in the caption for panel A lies with the authors' terminology. I was looking for a description or call out to particular pairs of residues or differences in interactions between each GP1-sAD complex that were different than other complexes within the asymmetric unit - not between each complex relative to the others. This panel was originally very small so it was hard to see the B-factor based tube width differences and exactly where they were located. Regardless, it would still be helpful to clarify with rewording or to state whether the differences lie in any meaningful places or if they merely exist on surface loops, etc.

comment 5:

5) For the MLV-based pseudovirion experiments, the methods indicate that the viruses are concentrated by 10x, however there is no mention of ensuring the amount of each virus delivered in these experiments is equal. One could certainly expect there to be variation in total titer and that this variation would prohibit comparison of the potency of Arenacept/sAD between each virus.

Indeed, the affinity of sAD for SABV is 10-fold greater than for JUNV, yet the neutralization potency is 2x lower for SABV. The

authors should at the very least normalize for the amount of virus delivered to the cells or comment on the disconnect between affinity and potency.

Reviewer #1 is absolutely correct that there are variations in the titers of the pseudoviruses that we use.

We are fully aware of these variations. However, the IC50 values are intrinsic property of the reagent and the particular virus that is used. That is to say, these IC50 values are titer-independent and the exact same value (within experimental error) should be obtained regardless of the titers used. The titers

of the viruses do affect the magnitude of the reporting signal that we record. All of our neutralization experiments were repeated multiple times, using different batches of viral stocks, presumably with different titers. However, as one can clearly see, the individually measured IC50 values (that we now include in a revised figure 4b) are nicely clustering, indicating high reproducibility. The fact that we always define maximal infectivity as 100%, allows us to effectively normalize all the different

experiments as reviewer #1 suggested us to do.

As for the observed affinities vs. neutralization potencies; mainly due to the effect of avidity, the large gaps in binding affinities (measured using monomeric sAD) are not preserved when comparing neutralization potencies (measured using a bivalent immunoadhesin). Also, although clearly related, neutralization and binding are two very different activities that are governed by very different kinetic

constants due to the native trimeric spikes on one hand vs. isolated GP1 domains on the other hand. Hence, we cannot assume that relative differences in binding affinities will be maintained when neutralization potencies are compared.

I still don't fully agree with their response. Yes, within a virus you can normalize to % total infection and their relatively tight error bars suggest that within each pseudovirion, the titers are about the same. But, unless they have somehow controlled for the amount of virus added for each particular pseudovirion, they can't compare IC50 values across viruses (but maybe they aren't trying to). Here, panel 4B looks like MACV is better than all the rest (which is supported by the SPR-derived affinity) and that the other three are about the same as one another (not necessarily correlated with affinity). You can imagine that if one is comparing neutralization of an antibody across a panel of different viruses that if one doesn't take measures to make sure a similar amount of virus is used, the potency for the Ab against that panel won't be comparative.

Anyway, if the singular point for this panel is that Areancept neuts these four viruses, fine. But as a reader, I immediately want to think it neuts MACV the best and neuts the other three about the same. Maybe this is true, maybe it's not - the reader can't tell because of the lack of detail from the authors as to how the experiments were done and controlled. It would be helpful for them to either point out that because absolute titer isn't controlled, that they can't make comparisons between the viruses (they do quantify virus using RT-PCR in the reference cited for pseudovirion production, so it is possible), or say something about using a particular amount of virus to achieve a particular % infection (which isn't perfect either since of course the different viruses have different affinities for the cell surface receptor and/or general differences in infectivity, but is perhaps better than nothing). I do note that whether they use the same amount of virus doesn't alter the message of "arenacept is broadly neutralizing", so it's probably fine as-is.

Reviewer #2 (Remarks to the Author):

The authors have answered all of my questions satisfactorily and have addressed all issues.

Reviewer #1 (Remarks to the Author):

comment 8:

8) Panel A in Figure 3 is too small to be meaningful. Can this panel be moved to a supplementary figure and made larger? Otherwise, it adds very little to the overall figure. Further, in the figure legend for this panel the authors say "there are differences in the protein pairs" but do not say what these differences are or if they are important.

Response: "As suggested by reviewer #1, we modified this panel to make it clearer. Also, please note that in the figure legend we do specify the differences, which relate to the B-factors of the different chains: "...There are differences between the protein pairs in the asymmetric unit; some are well defined in the electron density, having low B-factors (e.g., green / cyan), others are less defined and hence have higher B-factor (e.g., purple / orange)..."

The confusion in the caption for panel A lies with the authors' terminology. I was looking for a description or call out to particular pairs of residues or differences in interactions between each GP1-sAD complex that were different than other complexes within the asymmetric unit - not between each complex relative to the others. This panel was originally very small so it was hard to see the B-factor based tube width differences and exactly where they were located. Regardless, it would still be helpful to clarify with rewording or to state whether the differences lie in any meaningful places or if they merely exist on surface loops, etc.

We do not observe any meaningful differences between the protein pairs in the asymmetric unit. We now better clarify this point in the figure legend "...The protein pairs in the asymmetric unit differ in quality of the electron density; some have a low B-factors (e.g., green / cyan), while others are less defined and hence have higher B-factor (e.g., purple / orange). Regardless, the sAD/GP1 interface is identical in all pairs."

comment 5:

5) For the MLV-based pseudovirion experiments, the methods indicate that the viruses are concentrated by 10x, however there is no mention of ensuring the amount of each virus delivered in these experiments is equal. One could certainly expect there to be variation in total titer and that this variation would prohibit comparison of the potency of Arenacept/sAD between each virus. Indeed, the affinity of sAD for SABV is 10-fold greater than for JUNV, yet the neutralization potency is 2x lower for SABV. The authors should at the very least normalize for the amount of virus delivered to the cells or comment on the disconnect between affinity and potency.

Reviewer #1 is absolutely correct that there are variations in the titers of the pseudoviruses that we use. We are fully aware of these variations. However, the IC50 values are intrinsic property of the reagent and the particular virus that is used. That is to say, these IC50 values are titer-independent and the exact same value (within experimental error) should be obtained regardless of the titers used. The titers of the viruses do affect the magnitude of the reporting signal that we record. All of our neutralization experiments were repeated multiple times, using different batches of viral stocks, presumably with different titers. However, as one can clearly see, the individually measured IC50 values (that we now include in a revised figure 4b) are nicely clustering, indicating high reproducibility. The fact that we always define maximal infectivity as 100%, allows us to effectively normalize all the different experiments as reviewer #1 suggested us to do.

As for the observed affinities vs. neutralization potencies; mainly due to the effect of avidity, the large gaps in binding affinities (measured using monomeric sAD) are not preserved when comparing neutralization potencies (measured using a bivalent immunoadhesin). Also, although clearly related, neutralization and binding are two very different activities that are governed by very different kinetic constants due to the native trimeric spikes on one hand vs. isolated GP1 domains on the other hand. Hence, we cannot assume that relative differences in binding affinities will be maintained when neutralization potencies are compared.

I still don't fully agree with their response. Yes, within a virus you can normalize to % total infection and their relatively tight error bars suggest that within each pseudovirion, the titers are about the same. But, unless they have somehow controlled for the amount of virus added for each particular pseudovirion, they can't compare IC50 values across viruses (but maybe they aren't trying to). Here, panel 4B looks like MACV is better than all the rest (which is supported by the SPR-derived affinity) and that the other three are about the same as one another (not necessarily correlated with affinity). You can imagine that if one is comparing neutralization of an antibody across a panel of different viruses that if one doesn't take measures to make sure a similar amount of virus is

used, the potency for the Ab against that panel won't be comparative.

Anyway, if the singular point for this panel is that Arecept neuts these four viruses, fine. But as a reader, I immediately want to think it neuts MACV the best and neuts the other three about the same. Maybe this is true, maybe it's not - the reader can't tell because of the lack of detail from the authors as to how the experiments were done and controlled. It would be helpful for them to either point out that because absolute titer isn't controlled, that they can't make comparisons between the viruses (they do quantify virus using RT-PCR in the reference cited for pseudovirion production, so it is possible), or say something about using a particular amount of virus to achieve a particular % infection (which isn't perfect either since of course the different viruses have different affinities for the cell surface receptor and/or general differences in infectivity, but is perhaps better than nothing). I do note that whether they use the same amount of virus doesn't alter the message of "arencept is broadly neutralizing", so it's probably fine as-is.

As reviewer #1 commented, the main purpose of figure 4b is to demonstrate the ability of Arencept to neutralize a diverse set of viruses. Therefore, variations in the actual IC50 values (that all cluster to a very narrow set of values!) will not make any conceptual change to the results or to the conclusions of our manuscript. To be fully transparent about that, we now included a remark in the figure legend stating that we did not control the titers of pseudoviruses used in these experiments.

More generally, we kindly but strongly disagree with reviewer #1 that variations in titers would have an effect on the measured IC50 values, and will hence prevent a reader from comparing the potencies of Arencept against various GPCs that we used in the very same MLV-system. We already explained our reasoning in the previous round of revision. It is worth emphasizing that others have addressed this point in the past. A good reference would be the comprehensive validation of the TZM-bl assay by David Montefiori (Sarzotti-Kelsoe M. et al., J Immunol Methods, 2015). I am including below figure 3 from Montefiori's manuscript showing that substantial dilutions of the viral stocks result with very similar neutralization curves. The calculated IC50 values in these TZM-bl experiments had less than 3-fold differences despite the large differences in titers, which probably reflect the experimental error.

Moreover, reviewer #1 mentioned that we have used in the past RT-PCR to measure viral titers. This is true and we did that when the actual amounts of viruses were critical to the assay that we performed. In those experiments we intend to look at the maximal entry potentials of viruses using specific GPCs / mutant versions of the GPCs, which unlike IC50, are titer-dependent. Interestingly, from these past RT-PCR measurements that we performed in our lab (our own published and unpublished data), we know that the variation in titers that we get in our MLV system are limited. In all the measurements that we did before, and for various GPCs (i.e. belonging to different viruses), we never observed variations of more than 10-fold in the titers, and typically much less.

Reviewer #2 (Remarks to the Author):

The authors have answered all of my questions satisfactorily and have addressed all issues.

We thank reviewer #2.